# Gaze Beyond the Frame:
# Forecasting Egocentric 3D Visual Span

**Heeseung Yun**[1], **Joonil Na**[1], **Jaeyeon Kim**[2], **Calvin Murdock**[3], **Gunhee Kim**[1]

[1]Seoul National University, [2]Carnegie Mellon University, [3]Reality Labs Research at Meta

https://hs-yn.github.io/GazeBeyondFrame/

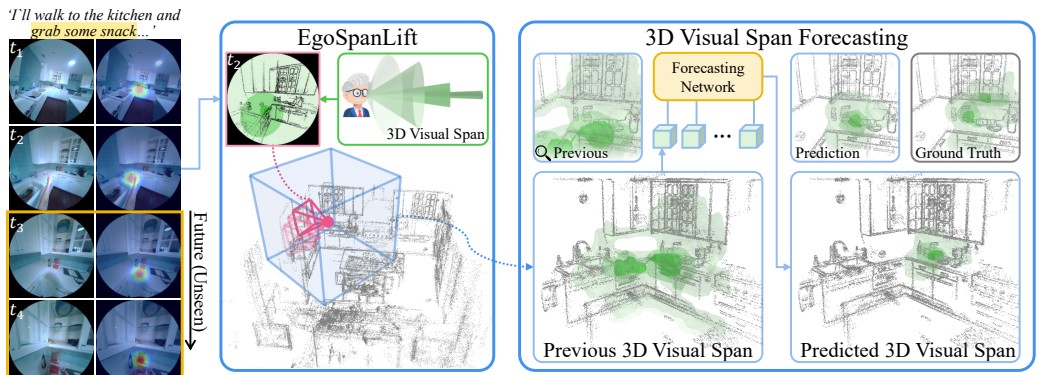

Figure 1: *Can we forecast our gaze beyond the frame?* We aim to predict a person's future visual focus in 3D surrounding environment by lifting egocentric 2D gaze history to 3D regions and forecasting future 3D visual spans from previous observations.

## Abstract

People continuously perceive and interact with their surroundings based on underlying intentions that drive their exploration and behaviors. While research in egocentric user and scene understanding has focused primarily on motion and contact-based interaction, forecasting human visual perception itself remains less explored despite its fundamental role in guiding human actions and its implications for AR/VR and assistive technologies. We address the challenge of egocentric 3D visual span forecasting, predicting where a person's visual perception will focus next within their three-dimensional environment. To this end, we propose EgoSpanLift, a novel method that transforms egocentric visual span forecasting from 2D image planes to 3D scenes. EgoSpanLift converts SLAM-derived keypoints into gaze-compatible geometry and extracts volumetric visual span regions. We further combine EgoSpanLift with 3D U-Net and unidirectional transformers, enabling spatio-temporal fusion to efficiently predict future visual span in the 3D grid. In addition, we curate a comprehensive benchmark from raw egocentric multisensory data, creating a testbed with 364.6K samples for 3D visual span forecasting. Our approach outperforms competitive baselines for egocentric 2D gaze anticipation and 3D localization while achieving comparable results even when projected back onto 2D image planes without additional 2D-specific training.

## 1 Introduction

People continuously perceive and interact with their surrounding environments in everyday life. Underlying these interactions are their intentions, which drive them to actively explore their surround-

39th Conference on Neural Information Processing Systems (NeurIPS 2025).

ings and engage in various behaviors. Understanding how individuals will perceive contexts and take actions in advance becomes a crucial element in comprehending their overall behavior patterns. The ability to accurately anticipate behavior not only reduces latency in real-time egocentric applications but also serves as a foundation for proactively delivering information and services in our daily lives, *e.g.*, immersive VR/AR, ambient computing, and assisting individuals with impairments.

To better understand a person's intentions, research in egocentric user and scene understanding has been widely explored, primarily involving action and contact-based interaction. For instance, when given egocentric visual or multimodal context, studies have focused on predicting subsequent human actions [1, 2, 3] or localizing regions in 2D images where interaction will occur [4]. Recent studies have actively investigated human pose or motion forecasting in 3D space [5, 6, 7], as well as contact location prediction during object interactions [8, 9]. However, attempts to forecast human visual perception itself remain less explored. Studies in vision science and cognitive psychology suggest that perceptual exploration significantly influences human motion [10, 11, 12, 13], and intuitively, most of our daily interactions follow perception, *i.e.*, we look before we leap. Therefore, forecasting visual perception is essential for proactively understanding and anticipating human behavior.

In this work, we address the novel challenge of *egocentric 3D visual span forecasting*–forecasting where a person's visual perception will be focused in the surrounding environment. We draw inspiration from the vision science literature regarding text reading [14, 15] and object/scene perception [16, 17, 18], where the visual span often refers to the fixated region of human vision from peripheral awareness to precise foveal gaze fixations. In our context, we adopt this term as *3D Visual Span* to refer to egocentric visual focus in 3D surroundings for addressing daily and casual behaviors and interactions. While previous research has shown impressive results in predicting egocentric future gaze fixations on 2D image frames [19, 20], forecasting gaze for dynamic scenarios in 2D remains unclear. Gaze anticipation requires jointly modeling the user's self-motion and attention–both of which are naturally directed toward specific locations in 3D space rather than arbitrary regions in 2D projections. That is, modeling visual span as 3D regions offers a more accurate and consistent representation of perceptual focus even beyond our current observations, unlocking promising egocentric applications that call for robust and proactive content rendering [21, 22].

Our main contribution comprises three key components: (i) a method for bridging egocentric visual spans and 3D scenes, (ii) a framework for forecasting future spans, and (iii) a newly curated benchmark from raw egocentric multisensory data. First, we propose *EgoSpanLift*, a novel method that lifts visual spans in 2D image frames into structured 3D volumetric regions, as illustrated in Fig. 1. Unlike existing 3D egocentric localization approaches on motion trajectories or object interactions, our method uniquely transforms SLAM-derived keypoints into gaze-compatible geometry to precisely extract volumetric regions corresponding to 3D visual span. *EgoSpanLift* encodes multi-level span information grounded in vision science [15]–ranging from wide head-orientation-based regions to fine-grained foveal fixations–enabling a semantically rich understanding of where and how people look in space. Building on this, we introduce an autoregressive forecasting framework that combines *EgoSpanLift* with a 3D U-Net for spatial representation using a unidirectional transformer for temporal modeling, effectively capturing the evolving dynamics of egocentric visual attention.

To establish a testbed at scale, we rigorously curate raw egocentric multisensory data streams [23, 24] for benchmarking 3D visual span forecasting under daily and skilled activity scenarios, namely FoVS-Aria and FoVS-EgoExo, covering a total of 364.6K samples. Our framework outperforms several competitive baselines, including frameworks for 3D egocentric localization and 2D gaze prediction models equipped with our EgoSpanLift. We conduct extensive analysis across varying spatio-temporal windows and activity categories. Notably, when our 3D visual span predictions are back-projected onto 2D image planes, they achieve accuracy on par with 2D-specific models even without 2D-specific training, demonstrating the versatility and robustness of modeling gaze in 3D.

## 2   Related Work

**Egocentric Anticipation of User Behavior.** Predicting user behavior from egocentric observations represents one of the core challenges in egocentric user and scene understanding. The EPIC-Kitchens dataset [1] introduced the challenge of predicting future action classes within procedural kitchen activities. Various approaches have been proposed to address this problem, including dual LSTM architectures for past and future modeling [2], and contrastive learning with RNNs for future visual

feature learning [3]. Beyond semantic action classification, studies have also proposed methods for predicting locomotion trajectories from current user observations [4, 25, 26]. For more detailed user posture prediction beyond trajectories [27], recent work has utilized reinforcement learning-based recurrent control [5], MLP-based residual modeling [6], and video-pose bimodal transformers [7].

**Egocentric Gaze Prediction** Understanding and predicting human gaze behavior is crucial for modeling various aspects of perception and interaction. Many studies primarily aim to estimate gaze direction from facial/head images [28, 29, 30], predict the object of visual attention [31, 32, 33, 34, 35], focus on VR/mobile scenarios [36, 37, 38, 39, 40], and model the interplay with language [41, 42]. Estimating and predicting gaze from egocentric perspectives pose additional challenges due to complex egocentric scenes and a combination of dynamic gaze shifts with frequent head and body movements, *i.e.*, self-motion. Egocentric gaze estimation often involves bottom-up saliency [43], joint gaze-action prediction [44], and global-local correlation [45].

On the other hand, the frontier of gaze anticipation remains relatively unexplored due to insufficient input context for predicting *future* frames. Zhang *et al.* [19] employ a GAN to synthesize future frames and forecast gaze locations, while Lai *et al.* [20] also introduce a multimodal anticipation model that uses contrastive spatial/temporal separable fusion to capture audio-visual correlations and improve future gaze prediction. However, predicting gaze in 2D images is often ill-posed in dynamic scenarios, as it requires jointly modeling the user's self-motion and attention, which are inherently directed toward specific locations in 3D rather than arbitrary regions in 2D frames. In this work, we address these issues by directly predicting the user's gaze and visual focus in 3D scenes through spatio-temporal fusion of volumetric representations.

**Egocentric 3D Interaction.** Research on egocentric interaction has evolved from 2D to 3D understanding and from contact-based to intention-based analysis. Earlier work on 2D image interaction analysis primarily addresses hand detection, segmentation, and pose estimation [46, 47, 48, 49, 50]. Moving beyond 2D approaches, 3D-based frameworks leverage hand poses and object interactions involving physical contacts from RGB(-D) inputs [51, 52, 53, 54, 55, 56, 57], allowing for precise prediction of hand trajectories and action targets [58, 59, 60]. Recent line of work interprets potential 3D interaction regions as spatial affordances, learned from either synthesized geometry and motion [9] or intentional cues in 2D interaction images [8].

Concurrent research with ours, FICTION [61], focuses on 4D human-object interaction. FICTION jointly predicts the 3D bounding box of objects involved in physical interactions along with the user's spatial location and body poses where contact occurs at future time steps. A key distinction of our work is that it address the forecasting problem for 3D regions where the user's visual perception is focused and precedes these everyday actions and interactions. Our predictions take the form of multi-level 3D volumetric regions of the visual span, enabling more fine-grained forecasting of potential interaction hotspots than bounding boxes.

## 3 EgoSpanLift: Lifting Egocentric Gaze Prediction from 2D to 3D

### 3.1 Preliminary: Simultaneous Localization and Mapping

Simultaneous Localization and Mapping (SLAM) refers to the problem of jointly estimating an agent's position and reconstructing the surrounding 3D environment from sequential observations. It is often formulated as a probabilistic framework that maximizes the posterior distribution $P(\mathbf{x}_t, \mathbf{m} \mid \mathbf{c}_{1..t-1}, \mathbf{o}_{1..t-1})$, where $\mathbf{x}_t$ denotes the pose at time $t$, $\mathbf{m}$ is the 3D mapping information, and $\mathbf{c}_{1..t-1}, \mathbf{o}_{1..t-1}$ are the control and observation history [62]. SLAM algorithms have been developed across sensing modalities, including LiDAR and RGB-D, with visual SLAM offering solutions based on feature matching [63], direct photometric tracking [64, 65], and visual-inertial fusion [66]. With growing efficiency and robustness, these methods output semidense 3D keypoints and time-aligned camera poses, providing structurally faithful and computationally efficient information about 3D scenes that users perceive and interact with from an egocentric perspective.

### 3.2 Keypoint Selection and Classification

**Observation-based Keypoint Selection.** An overview of EgoSpanLift is illustrated in Fig. 2. Our method initiates with a set of 3D semidense keypoints $\mathcal{P}$ and localization information $\mathcal{E}$, typically obtained through visual SLAM [67]. Specifically, each $p_i \in \mathcal{P}$ consists of $(\mathbf{p}_i, \sigma_i, t_i)$, where $\mathbf{p}_i \in \mathbb{R}^3$

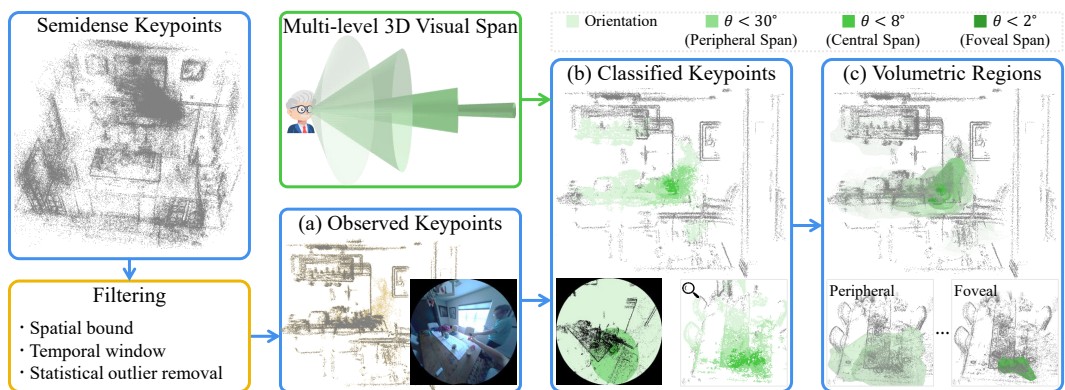

Figure 2: Overview of EgoSpanLift. Using 3D semidense keypoints from egocentric observations, *e.g.*, SLAM, we filter observed keypoints at a given time window and leverage multi-level human visual span to compute volumetric regions in 3D scenes.

is the 3D keypoint in global coordinates, $\sigma_i$ represents the confidence of estimation in the form of the variance of inverse distance, and $t_i$ denotes the time at which the point was observed. Additionally, $\mathbf{E}_t \in \mathcal{E}$ represents an SE(3) element that transforms from the local egocentric frame (*i.e.*, central pupil frame) at time $t$ to the global coordinate system, *i.e.*, $\mathbf{E}_t = \begin{pmatrix} \mathbf{R}_t & \mathbf{t}_t \\ \mathbf{0}^T & 1 \end{pmatrix}$, where the $z$-axis in the local coordinates denotes forward. From this, we obtain the set of keypoints observed at time $t$:

$$\mathcal{P}_t = \{p_i \in \mathcal{P} \mid t_i = t, \|\mathbf{p}_i - \mathbf{t}_t\|_1 < D/2, \mathcal{I}_f(p_i; \mathcal{P}_t) = 1\}, \tag{1}$$

where the first term serves as a filter that preserves visually observed keypoints at the given time, and the second term acts as a spatial filter that retains only points within a cubic boundary of length $D$ around the user. The last term, $\mathcal{I}_f(p_i; \mathcal{P})$, is an indicator function returning the result of neighbor-based statistical outlier filtering $f$, where 1 indicates validity and 0 indicates invalidity.

We can establish the rationale for each filtering term as follows. For temporal filtering, we consider only a limited context within a temporal window spanning a few seconds before the current time rather than all keypoints observed from beginning to end. This enables operation without requiring an offline algorithm and prevents the inclusion of points that were visible from different viewpoints but are currently occluded (*e.g.*, items inside the fridge). For spatial filtering, we restrict the area to prevent gaze from erroneously overshooting to distant locations and to focus on regions within the user's vicinity that will be effectively perceived and interacted with. In most experiments, we used $D = 3.2$m, though we discuss experiments extending beyond 6m in Sec. 5.2. For outlier removal, we use neighbor-based statistical filtering instead of confidence-based filtering, *i.e.*, $\sigma_i$, which is typically used when extracting static 3D scenes from SLAM outputs. This approach can retain important dynamic elements in egocentric scenes, such as moving people, the user's hands, and other dynamic objects in the space, while effectively removing invalid points.

**Gaze-based Keypoint Classification.** Using keypoints from $\mathcal{P}_t$, we classify points that fall within a visual span defined around the user's gaze direction after transforming each point to the local coordinates as $\mathbf{E}_t^{-1}\mathbf{p}_i$. To determine whether points are included in the visual span, we utilize a 3D gaze representation defined as a cone extending from the user. While some previous works have employed egocentric 3D representations in the form of directions or cones, *e.g.*, intersection between gaze vectors and triangulated meshes of static objects [68] or overlap of multiple users' gaze cones [69], our approach features a key distinction as it covers more general use cases by addressing which local regions of a 3D scene capture an *individual* user's visual attention during daily activities without discriminating between *dynamic* and static components.

When the aforementioned gaze cone and keypoints at the local coordinates are projected onto the $z = 1$ plane, they are represented as green ellipses and black dots in the lower left of Fig. 2-(b). We select the dots that fall within these ellipses using the angular distance threshold $\theta$ (*i.e.*, eccentricity):

$$Q_t^{\theta, \mathbf{g}_t} = \left\{ p_i \in \mathcal{P}_t \; \middle| \; \frac{< \mathbf{E}_t^{-1}\mathbf{p}_i, \mathbf{g}_t >}{\|\mathbf{E}_t^{-1}\mathbf{p}_i\|\|\mathbf{g}_t\|} > \cos\theta \right\}, \tag{2}$$

where $< \cdot, \cdot >$ denotes inner product, and $\mathbf{g}_t$ denotes the gaze direction in a local frame at time $t$. Through this, we can classify a set of points that correspond to the egocentric 3D visual span.

While employing SLAM for visual spans may pose challenges in modeling regions between semidense keypoints, our approach remains effective for two main reasons. First, human visual span or peripheral vision is known to be significantly influenced by contrast as well as eccentricity from the gaze direction [70, 71]. Since people overwhelmingly interact with visually salient objects rather than empty white walls, this approach is highly compatible with keypoints obtained through SLAM. Indeed, >99% of the 3D visual spans in our curated data samples from raw egocentric multisensory observations overlapped with SLAM keypoints. Furthermore, rather than relying on keypoints as inputs or outputs, we represent visual spans as volumetric regions derived from them, *e.g.*, Eq. 3, which provides good coverage of the given 3D scene and helps mitigate the aforementioned challenges.

### 3.3 Multi-level Volumetric Region Localization

Using classified keypoints $Q_t^{\theta, g_t}$ in a cube of length $D$, we obtain regions corresponding to visual spans by computing their occupancy in a 3D Cartesian grid with resolution $R$ and duration $[t_b, t_e]$:

$$V_{[t_b, t_e]}^{\theta, \mathbf{g}_t}(i, j, k) = \mathcal{I}(\left|\{p_i \in \cup_{t \in [t_b, t_e]} Q_t^{\theta, \mathbf{g}_t} | \mathbf{0} \le (p_i - \mathbf{t}_{t_b} + D/2) \times R/D - (i, j, k) \le \mathbf{1}\}\right| > 0), \quad (3)$$

where $\mathcal{I}$ is an indicator function and $|\cdot|$ represents the cardinality of a set. This approach indirectly covers the regions not captured by semidense keypoints while maintaining constant space complexity regardless of the increasing number of points with longer durations. Also, since $V$ is represented as a binary occupancy grid, computing the similarity between overlapped visual spans can be trivial.

Inspired by taxonomy defined in vision science literature [15], we categorize 3D volumetric regions of vision spans into four levels, as illustrated in Fig. 2. First, foveal localization $V^{\theta_f, \mathbf{g}_t}$ corresponds to the conventional 2D gaze localization area, covering a highly localized region with an eccentricity of $\theta_f = 2°$. While foveal span exists in most regions, *i.e.*, around 80% in our curated dataset, there may be instances where it is absent due to our use of semidense keypoints. To address this limitation, we utilize spans corresponding to wider eccentricities: the central span $V^{\theta_c, \mathbf{g}_t}$ with $\theta_c = 8°$ and the near peripheral span $V^{\theta_p, \mathbf{g}_t}$ with $\theta_p = 30°$. Finally, regions observed in spans defined beyond these are significantly influenced by head orientation as much as the gaze, with a complex interplay between the two [72, 73]. Therefore, we employ the most broadly defined span as the region within the field-of-view centered on head orientation, *i.e.*, $V^{\theta_o, \mathbf{z}}$, which is analogous to the view frustum capturing visual input in 2D setups (*e.g.*, $\theta_o = 55°$ in our experiments). Note that while continuous distributions such as Gaussian could be used, we analyze through the lens of this well-established taxonomy for intuitive evaluation and level-by-level integration across multiple timesteps.

As a result, for a given temporal window, we obtain multi-level volumetric regions that represent varying degrees of user visual attention focus in the surrounding environment. Since EgoSpanLift does not rely on time-consuming algorithms, when combined with existing platforms capable of real-time SLAM and gaze processing [74, 75, 76], it enables the acquisition of 3D information about human visual attention with trivial latency. Additionally, its representation as points or volumetric regions in a 3D Cartesian grid facilitates adaptation to existing 3D frameworks.

## 4 Forecasting Network

One of the most representative problems when considering a user's visual span in 3D involves predicting future visual attention based on past contextual focus patterns. To this end, we introduce a straightforward framework to forecast 3D visual spans for future $T_f$ frames, given the history of observations from past $T_p$ frames, as illustrated in Fig. 3-(a). Note that our primary interest lies not in precisely matching what was seen in each respective frame but rather in understanding the trends and coverage areas of the user's attention over a given period. To perform precise evaluation while minimizing the effects of exploration order of gaze, we use the union of visual spans over $T_f$ frames as our prediction target.

**Autoregressive Encoder.** Given localization information $\mathcal{E}_{\text{prev}}$ and observed semidense keypoints $\mathcal{P}_{\text{prev}}$ for the past $T_p$ seconds, we obtain multi-level volumetric regions representing previous visual spans through EgoSpanLift as described in Sec. 3. Specifically, we utilize a grid of size $T_p \times (4 + 1) \times R \times R \times R$ as input, where 4 represents visual spans ranging from orientation to foveal span,

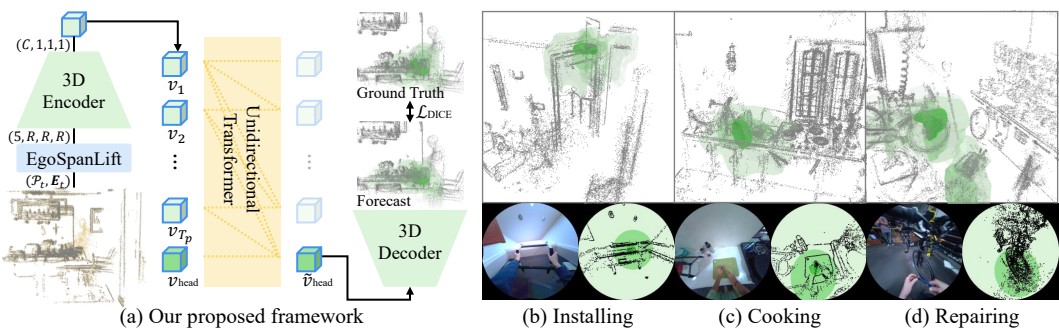

Figure 3: (a) Illustration of our framework and (b-d) diverse scenario examples in our curated dataset.

and 1 represents the complete scene in the given space regardless of span inclusion. To extract spatial features from this input grid, we initially compress them using the encoder of a 3D U-Net [77]. By reducing the spatial dimension by a factor of $R$ through pooling, we obtain encoded features of size $T_p \times C$, where $C$ is the feature dimension. Note that performing spatial reduction by a factor smaller than $R$ showed negligible impact on performance.

While using $T_p$ embeddings encodes how visual span has evolved over time, we separately incorporate a global embedding to serve as a prediction head. Thus, we utilize an embedding that encodes all visual spans within the duration as a global embedding, serving as our prediction head. By appending this embedding after the other embeddings, we obtain features of size $(T_p + 1) \times C$, $i.e.$, $v_1, ..., v_{T_p}, v_{\text{head}}$. To learn temporal dependencies among these features and increase the model's expressiveness, we employ a transformer with a unidirectional attention mask [78, 79], ensuring that information about temporal dynamics is integrated toward the final global embedding.

**Decoder.** Using the output embedding $\tilde{v}_{\text{head}}$ that corresponds to $v_{\text{head}}$ from the transformer, we perform upsampling via a U-Net decoder. During this process, we utilize intermediary features from the U-Net encoder through residual connections. By applying sigmoid, we ultimately obtain an output $\tilde{Y}_{ijk}$ of size $4 \times R \times R \times R$ representing a 0-1 soft occupancy, which corresponds to our aggregated forecast of the user's visual span over the future $T_f$ frames.

**Learning Objective.** Since visual spans occupy only a very narrow region of the space surrounding the user, learning meaningful signals through conventional cross-entropy losses could be challenging. For instance, in the case of foveal span, almost all samples occupy less than 1% of the entire grid region. Therefore, we train our model using the dice loss with the ground truth forecast $Y_{ijk}$, where $\odot$ is elementwise product:

$$\mathcal{L} = 1 - \frac{2 \times \sum \tilde{Y}_{ijk} \odot Y_{ijk}}{\sum \tilde{Y}_{ijk} + \sum Y_{ijk} + 1}. \tag{4}$$

**Latency Analysis.** Our framework has two primary sources of latency: (i) extracting the relevant set of points from gaze and SLAM keypoints (performed every 100ms), and (ii) performing model inference every second on a set of points spanning two seconds. Since the point extraction can be pre-computed and stored at 10 fps for continuous use, we only need to consider the computation time for processing the final observation when calculating inference latency. We measured this in a resource-constrained environ-ment compared to our training setup, using 8 CPU cores and a GPU with 12GB VRAM.

Table 1: Latency analysis of our framework.

| Operation | Latency |
|---|---|
| Point preprocessing | 4.541±1.999ms |
| 3D visual span localization | 1.811±0.824ms |
| Voxelization | 45.406±26.223ms |
| Model inference | 19.483±8.234ms |
| **Average latency** | 71.241ms |

The results are summarized in Table 1. The first stage, which handles outlier removal, axis-aligned bounding box cropping for keypoints, and selection of points within a certain degree of eccentricity from the gaze, can be processed within 10ms. In the second stage, we identified that the primary bottleneck lies in voxelization rather than in the model itself. This occurs because the large number of keypoints from the previous stage should be voxelized, whereas the model operates efficiently once it receives the 3D voxelized representation. Consequently, the average inference latency is 71.241ms, yielding a real-time factor of 0.036, confirming our claim on fast processing capability. However, actual AR/VR environments typically operate with even fewer computational resources, and thus

Table 2: Comparison of forecasting accuracy on the FoVS-Aria test split. Higher the better.

| Methods | Orientation | | Peripheral$_{30°}$ | | Central$_{8°}$ | | Foveal$_{2°}$ | |
|---|---|---|---|---|---|---|---|---|
| | IoU | F1 | IoU | F1 | IoU | F1 | IoU | F1 |
| 2D Center Prior + EgoSpanLift + [80] | - | - | 0.4061 | 0.5583 | 0.1621 | 0.2497 | 0.0685 | 0.1049 |
| GLC [45] + EgoSpanLift + [80] | - | - | 0.4553 | 0.6064 | 0.2227 | 0.3267 | 0.1321 | 0.1873 |
| CSTS [20] + EgoSpanLift + [80] | - | - | 0.4567 | 0.6076 | 0.2342 | 0.3423 | 0.1388 | 0.1948 |
| OccFormer [81] | 0.1395 | 0.2352 | 0.1106 | 0.1846 | 0.0429 | 0.0632 | 0.0158 | 0.0289 |
| VoxFormer [82] | 0.2192 | 0.3093 | 0.1847 | 0.2580 | 0.0915 | 0.1279 | 0.0453 | 0.0704 |
| IAG [8] | 0.3250 | 0.3669 | 0.2154 | 0.2542 | 0.1250 | 0.1851 | 0.0747 | 0.1050 |
| EgoChoir [9] | 0.4959 | 0.6579 | 0.4302 | 0.5581 | 0.2612 | 0.3608 | 0.1987 | 0.2311 |
| Global Prior | 0.1359 | 0.2314 | 0.1048 | 0.1825 | 0.0331 | 0.0618 | 0.0146 | 0.0280 |
| Ours (w/o previous span) | 0.3424 | 0.4906 | 0.2616 | 0.3928 | 0.1071 | 0.1739 | 0.0594 | 0.0828 |
| Ours ($\mathcal{L}_{BCE}$) | 0.5730 | 0.7134 | 0.4594 | 0.6017 | 0.2836 | 0.3879 | 0.2059 | 0.2609 |
| Ours (Gaze-only + [80]) | - | - | 0.4723 | 0.6214 | 0.2666 | 0.3791 | 0.2494 | 0.3193 |
| Ours (Single-task) | 0.5832 | 0.7238 | 0.4721 | 0.6154 | 0.3351 | 0.4485 | 0.2494 | 0.3193 |
| Ours (w/o global embedding) | 0.5602 | 0.7042 | 0.4647 | 0.6128 | 0.3241 | 0.4476 | 0.2624 | 0.3487 |
| **Ours** (full) | **0.5838** | **0.7247** | **0.4886** | **0.6350** | **0.3513** | **0.4762** | **0.2836** | **0.3709** |

additional optimization techniques such as model quantization or more efficient voxelization could be considered for further performance improvements.

## 5 Benchmarking Egocentric 3D Visual Span Forecasting

### 5.1 Forecasting Egocentric Daily Activities

**Curation of FoVS-Aria.** Due to the lack of an existing testbed for egocentric 3D visual span forecasting, we curate a dataset by processing the raw data streams from an existing egocentric multisensory dataset. Aria Everyday Activities dataset [23] encompasses diverse scenarios of people engaging in daily activities across different environments and interacting with others, comprising 143 recordings with a total duration of 7.3 hours. Initially, we inspect the SLAM keypoints and gaze scenarios of all recordings, manually filtering out cases with insufficient keypoints or those limited to stationary viewing of phones/TVs. Following recent research in 2D gaze anticipation [20], we define a sample as a 2-second prediction task based on a 2-second previous observation with a sliding window of 1 second. For spatial parameters, we use resolution $R = 16$ and cube length $D = 3.2$m, indicating that accurately matching a cell corresponds to precision within a $D/R = 20$cm error margin. We construct the test split using all recordings from location 4 to enable the evaluation of unseen locations and the validation split by randomly stratifying the rest. Consequently, our constructed FoVS(Forecasting 3D Visual Span)-Aria consists of 23.2k samples in total, with 19.3k, 1.9k, and 2.1k samples for train, validation, and test splits, respectively.

**Evaluation Protocol.** For multi-level 3D visual span, we evaluate each level separately. We primarily utilize 3D IoU as a primary metric since our main concern is the overlap of volumetric regions. Additionally, we report F1 scores, commonly used in 2D gaze evaluation [45, 20], while precision and recall are reported in the Appendix. Finally, since the foveal span is defined within a highly narrow region, overlap-based metrics cannot fully capture its distribution. Therefore, we also examine the distribution of the (metric) distance between the ground truth region and the predicted region. Analysis on saliency-based metrics is deferred to Appendix.

Regarding baselines, due to lack of well-established frameworks for our task configuration, we construct applicable models from various domains. First, we adapt 3D localization methods like OccFormer [81] and VoxFormer [82] to predict visual span, which are known for effectively inferring semantic labels for voxels in outdoor driving scenarios. We also employ IAG [8] and EgoChoir [9], models designed to predict affordances or interaction hotspots in 3D from an egocentric user-object interaction perspective. Another framework includes state-of-the-art methods for 2D gaze anticipation, such as GLC [45] and CSTS [20]. Thanks to our EgoSpanLift, we can project these models' 2D inference results to 3D, extrapolating head pose information from past context using a multi-task Gaussian process model [80].

**Comparison with Prior Arts.** Table 2 presents comparative results among various methods. Overall, there is a substantial performance gap between existing methods and our approach, with the disparity becoming more pronounced when predicting more localized regions in 3D (*e.g.*, foveal span), where our method outperforms others by more than 50%. EgoChoir [9] displays the most promising performance among baselines, due to its capability of predicting 3D interaction hotspots that could be in line with user intentions. Compared to the other 3D localization methods, frameworks that consider the gaze dynamics show improved performance, despite falling significantly short on foveal span forecasting. This performance gap is further evidenced in Table 3, where baseline frameworks produce errors approximately twice as high as our approach on average.

Table 3: Distribution of metric errors in 3D foveal span localization.

| Distance (cm) | min. | avg. | max. |
|---|---|---|---|
| GLC [45] | 59.65 | 73.47 | 87.20 |
| CSTS [20] | 59.71 | 73.79 | 87.68 |
| w/o prev. span | 66.94 | 83.07 | 98.98 |
| Single-task | 36.54 | 52.90 | 69.18 |
| **Ours** | **19.04** | **34.85** | **51.23** |

**Ablation Studies.** The last six rows in Table 2 display the results of ablation studies on our methodology. The global prior—predicting answers based on the forecast distribution of the train split—yields considerably low numbers, implying that our FoVS-Aria encompasses diverse visual spans driven by various user intentions. Additionally, our model's performance significantly deteriorates when attempting prediction without previous span information; this indicates that knowledge of where the user previously focused is crucial for accurate forecasting, as visual span can vary substantially according to intention. Using binary cross-entropy (BCE) as a loss function results in marginal performance degradation up to the central span level, but shows marked decline in foveal span performance. Similarly, jointly solving multi-level spans consistently outperforms the single-task approach, particularly benefiting foveal span improvement.

Multi-level interpretation carries significant implications due to uncertainties in self-motion forecasting and geometric correspondence between 2D and 3D spaces, as exemplified in suboptimal performance of predicting only gaze in 2D or 3D and applying postprocessing [80]. Two key conceptual differences between our framework and the postprocessing variant are (i) learning from multi-level representation and (ii) mitigating uncertainties in self-motion anticipation through end-to-end prediction. Leveraging the interconnection between gaze and periphery [15] allows us to capture cues about future gaze from previous periphery and forecast future periphery in light of previous gaze, which can be observed in several qualitative examples. Moreover, given the nature of the viewing frustum, extending 3D gaze predictions to broader spans necessitates the forecasting of egocentric 6DoF pose trajectories with a separate postprocessing stage, which propagates uncertainties in self-motion forecasting and geometric correspondence between 2D and 3D spaces. In contrast, our end-to-end framework does not assume a specific forecast trajectory and predicts plausible 3D spans within the scene while implicitly learning to mitigate such uncertainties.

## 5.2 Forecasting Skilled Activites at Scale

**Curation of FoVS-EgoExo.** To analyze the effectiveness of visual span forecasting in skilled activities at a larger scale, we utilize Ego-Exo4D [24] as our source data, comprising hundreds of hours of observations. Ego-Exo4D consists of eight activity categories, from which we excluded soccer, basketball, and dance, as these activities involve overly large or dynamic scenes where clear fixations in visual spans are unidentifiable in the majority of cases. Instead, we focus on five categories: cooking, music, health, repair, and bouldering. After collecting only the footage corresponding to actual task execution, we refine the data and calculate volumetric regions following the same procedure used for FoVS-Aria. Considering the increased scale and the focus on specific tasks, we establish a 4-second prediction as the default setting. Thus, we collect a total of 341.4k samples, divided into 274.7k, 29.6k, and 37.0k samples for train, validation, and test splits, respectively.

**Performance Analysis.** Table 4 compares baseline model performance on FoVS-EgoExo. While our approach shows substantial performance gains over FoVS-Aria, presumably due to the nature of skilled activities and an order of magnitude larger sample size, the corresponding improvements in baseline methods are relatively modest. Unlike FoVS-Aria, the change of numbers regarding the orientation span likely stems from the inherent characteristics of skilled activities in FoVS-EgoExo, where participants tend to maintain more sustained focus on specific tasks rather than engaging in frequent, diverse head movements. Given that visual span forecasting prioritizes accurate prediction

Table 4: Comparison of forecasting accuracy on the FoVS-EgoExo test split. Higher the better.

| Methods | Orientation | | Peripheral$_{30°}$ | | Central$_{8°}$ | | Foveal$_{2°}$ | |
|---|---|---|---|---|---|---|---|---|
| | IoU | F1 | IoU | F1 | IoU | F1 | IoU | F1 |
| CSTS [20] + EgoSpanLift + [80] | - | - | 0.4978 | 0.6398 | 0.2867 | 0.4010 | 0.1556 | 0.2107 |
| OccFormer [81] | 0.1287 | 0.2280 | 0.0920 | 0.1685 | 0.0251 | 0.0490 | 0.0084 | 0.0167 |
| VoxFormer [82] | 0.1896 | 0.3188 | 0.1475 | 0.2571 | 0.0620 | 0.1168 | 0.0179 | 0.0350 |
| EgoChoir [9] | 0.3287 | 0.4948 | 0.2851 | 0.4437 | 0.1976 | 0.3300 | 0.1266 | 0.2247 |
| Global Prior | 0.2329 | 0.3621 | 0.2478 | 0.3776 | 0.1245 | 0.2119 | 0.0658 | 0.1188 |
| Ours (w/o prev. span) | 0.4091 | 0.5665 | 0.3876 | 0.5296 | 0.2855 | 0.4107 | 0.2255 | 0.3152 |
| Ours ($\mathcal{L}_{BCE}$) | 0.5112 | 0.6621 | 0.4905 | 0.6338 | 0.3722 | 0.4867 | 0.2870 | 0.3578 |
| Ours (w/o global embedding) | 0.4998 | 0.6542 | 0.4892 | 0.6381 | 0.3954 | 0.5294 | 0.3475 | 0.4500 |
| **Ours** (full) | 0.5230 | 0.6743 | 0.5108 | 0.6569 | 0.4212 | 0.5541 | 0.3692 | 0.4702 |
| $D = 3.2, R = 32$ (10cm) | 0.4443 | 0.6040 | 0.4362 | 0.5893 | 0.3319 | 0.4656 | 0.2493 | 0.3497 |
| $D = 6.4, R = 32$ (20cm) | 0.4920 | 0.6462 | 0.4902 | 0.6394 | 0.4121 | 0.5464 | 0.3640 | 0.4689 |
| $D = 3.2, R = 16$ (20cm) | 0.5230 | 0.6743 | 0.5108 | 0.6569 | 0.4212 | 0.5541 | 0.3692 | 0.4702 |

(a) Cross-category transfer of peripheral/foveal forecast (IoU)  (b) Influence of temporal window length

Figure 4: Analysis of our proposed framework on the FoVS-EgoExo test split.

of narrower regions, it is particularly noteworthy that substantial performance gaps exist between our method and baselines for Central and Foveal areas. These results indicate that skilled activity forecasting in FoVS-EgoExo is not inherently easier than daily activity forecasting in FoVS-Aria, while demonstrating that our model achieves robust performance across multiple scales and domains compared to prior arts.

We further analyze the impact of scale, category-specific training, and spatio-temporal granularity. According to experiments conducted with varying spatial configurations in Table 4, the impact on performance is marginal if the grid range is proportionally increased to maintain the same precision of 20cm despite an increased resolution of $R = 32$. However, for our model trained at 10cm precision with $R = 32$, performance decreases to levels similar to those observed in FoVS-Aria. Similarly, as demonstrated in Fig. 4-(b), performance tends to decline as we attempt to cover visual span further into the future, which is particularly significant when predicting foveal span.

Finally, cross-category transfer results are summarized in Fig. 4-(a). Despite significant variations in patterns across task categories, training an integrated model at sufficient scale achieves virtually identical performance to maintaining separate models for each category. Transfer among static tasks (*e.g.*, cooking, music, and health) is relatively effective, but less so for more dynamic tasks such as repair or bouldering. Additionally, peripheral span, which covers wider areas or visual exploration, preserves performance better in cross-category scenarios compared to foveal span, which targets more localized regions. Qualitative examples can be found in Fig. 5-(b) and the Appendix.

## 5.3 Extension to 2D Gaze Anticipation

For completeness in our experimental analysis, we explore whether the reverse direction—from 3D to 2D—is also possible using our framework, given that we successfully extended egocentric visual span from 2D to 3D. To this end, we utilize samples from FoVS-Aria to compare 2D gaze anticipation performance. While our previous experiments in Table 2 required separate procedures like EgoSpanLift to bring 2D gaze models into 3D, deflating our model's inference results to 2D can be accomplished trivially. We select the cell with maximum logit value from our model's foveal span prediction and project it onto the 2D image plane using the user's current head pose. Through

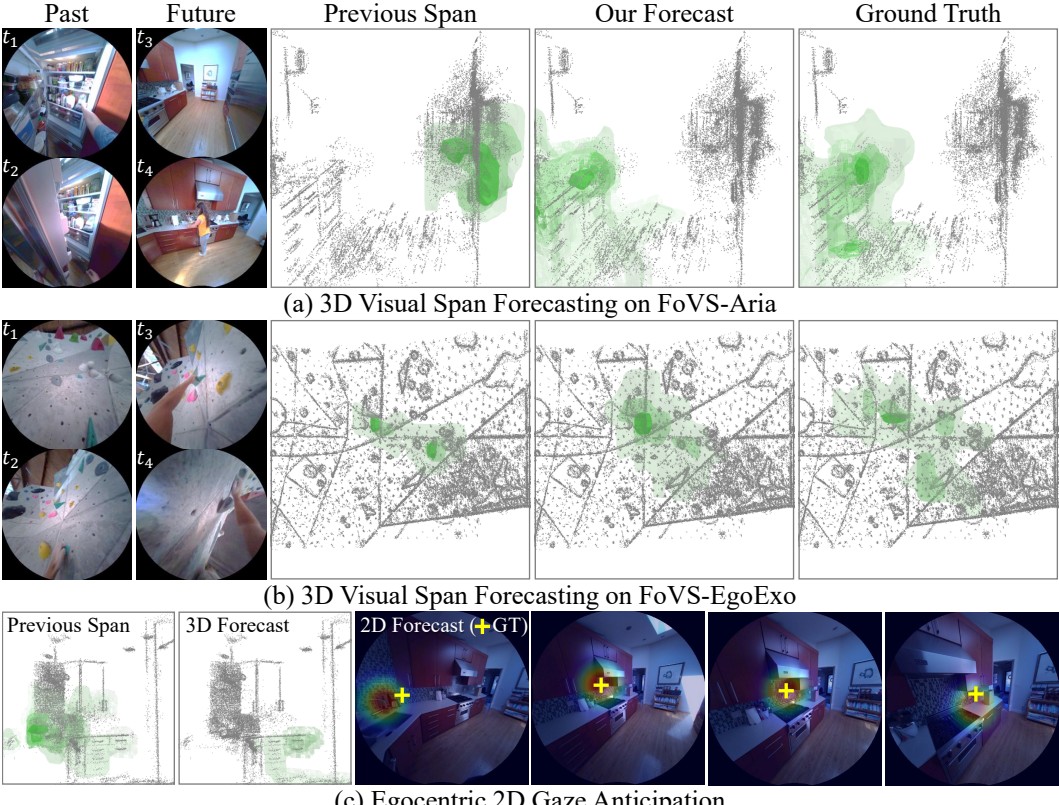

| Past | Future | Previous Span | Our Forecast | Ground Truth |

(a) 3D Visual Span Forecasting on FoVS-Aria

(b) 3D Visual Span Forecasting on FoVS-EgoExo

(c) Egocentric 2D Gaze Anticipation

Figure 5: Qualitative examples. Our framework effectively forecasts various scenarios, such as (a) closing the fridge and turning around or (b) deciding which rocks to grab and navigate.

interpolation between the user's current and projected gaze, we can easily obtain gaze anticipation results for the future two seconds.

The performance comparison results can be found in Table 5. Interestingly, while 2D methodologies struggled to accurately predict foveal span when transferred to 3D, our approach achieves on-par performance to models trained specifically for 2D despite not conducting any 2D-specific training. The baseline performance we obtained is slightly lower than that in the referenced paper [20], which we attribute to our exclusion of trivial recordings from FoVS-Aria like watching TV or smartphones.

Table 5: Comparison of egocentric 2D gaze anticipation.

|  | F1 | Pr | Re |
|---|---|---|---|
| GLC [45] | 0.505 | 0.453 | 0.571 |
| CSTS [20] | **0.515** | **0.497** | 0.535 |
| Ours (Single) | 0.505 | 0.432 | 0.608 |
| Ours | **0.515** | 0.440 | **0.619** |

## 6 Conclusion

We addressed the challenge of egocentric 3D visual span forecasting by designing a method that lifts multi-level visual span from 2D to 3D semidense keypoints and introducing an end-to-end framework for forecasting volumetric regions corresponding to each visual span. Furthermore, we curated a testbed for this problem by processing two datasets with raw multisensory observations, where we consistently outperformed a wide range of prior arts across all metrics. We anticipate that this framework will play a crucial role in proactively understanding human intent captured through perception that precedes interaction, enabling preemptive delivery of various latency-sensitive services. Future extensions of this work could incorporate forecasting problems for non-visual perception, such as auditory or proprioceptive inputs, or focus on forecasting highly precise attention tracking on dense scenes, which would represent particularly intriguing directions for further research.

**Acknowledgment.** This work was supported by Samsung Advanced Institute of Technology, Institute of Information & communications Technology Planning & Evaluation (IITP) grant funded by the Korea government (MSIT) (No. RS-2021-II211343, No. RS-2022-II220156, No. RS-2025-25442338), and National Research Foundation of Korea(NRF) funded by the Ministry of Education(RS-2023-00274280). This research was conducted as part of the Sovereign AI Foundation Model Project(Data Track), organized by the Ministry of Science and ICT(MSIT) and supported by the National Information Society Agency(NIA), S.Korea. Gunhee Kim is the corresponding author.

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

Figure 6: More examples from our curated FoVS-Aria and FoVS-EgoExo. Blue denotes previous observations and green denotes forecast targets.

# A  Experimental Details

## A.1  Dataset Curation Details

The detailed preprocessing procedures for FoVS-Aria and FoVS-EgoExo are as follows. We first temporally synchronize the raw streams present in each source dataset, namely gaze, RGB, audio, IMU-based trajectory, and SLAM observations. Since gaze data has the lowest sampling rate of 10Hz, we use it as the temporal anchor to synchronize other sensory streams and group them into 1-second units. Using a 1-second sliding window, we apply EgoSpanLift to obtain volumetric grids corresponding to multi-level visual spans. Given the substantial volume of raw data (*e.g.*, Ego-Exo4D exceeding 10TB), we store the processed data as bitpacked arrays to enable efficient storage and retrieval during training.

We perform manual validation for both datasets regarding gaze distribution and 3D point cloud integrity. For each recording, we visualize gaze distribution across 2D frames throughout the entire recording, excluding samples with extremely low variance as these indicated miscalibration. We also manually remove cases where SLAM capture quality is poor and visual spans with gaze fixations are trivial, such as instances of lying down while reading books or using mobile phones. Additionally, we exclude samples where keypoints of interest fall below two standard deviations from the dataset-wide

Table 5: Comparison of forecasting accuracy on the FoVS-Aria test split.

| Methods | Orientation | | | | Peripheral$_{30°}$ | | | | Central$_{8°}$ | | | | Foveal$_{2°}$ | | | |
|---|---|---|---|---|---|---|---|---|---|---|---|---|---|---|---|---|
| | IoU | F1 | Pr | Re | IoU | F1 | Pr | Re | IoU | F1 | Pr | Re | IoU | F1 | Pr | Re |
| 2D Center Prior + EgoSpanLift + [6] | 0.5868 | 0.7311 | 0.6513 | 0.8817 | 0.4061 | 0.5583 | 0.5570 | 0.6251 | 0.1621 | 0.2497 | 0.2704 | 0.2968 | 0.0685 | 0.1049 | 0.1132 | 0.1286 |
| GLC [7] + EgoSpanLift + [6] | 0.5868 | 0.7311 | 0.6513 | 0.8817 | 0.4553 | 0.6064 | 0.5880 | 0.6909 | 0.2227 | 0.3267 | 0.3569 | 0.3634 | 0.1321 | 0.1873 | 0.1962 | 0.2225 |
| CSTS [8] + EgoSpanLift + [6] | 0.5868 | 0.7311 | 0.6513 | 0.8817 | 0.4567 | 0.6076 | 0.5920 | 0.6909 | 0.2342 | 0.3423 | 0.3689 | 0.3865 | 0.1388 | 0.1948 | 0.2059 | 0.2295 |
| OccFormer [9] | 0.1395 | 0.2352 | 0.2340 | 0.2422 | 0.1106 | 0.1846 | 0.1799 | 0.1981 | 0.0429 | 0.0632 | 0.0916 | 0.0534 | 0.0158 | 0.0289 | 0.0313 | 0.0276 |
| VoxFormer [10] | 0.2192 | 0.3093 | 0.3544 | 0.3027 | 0.1847 | 0.2580 | 0.3600 | 0.2225 | 0.0915 | 0.1279 | 0.0943 | 0.2571 | 0.0453 | 0.0704 | 0.0541 | 0.1175 |
| IAG [11] | 0.3250 | 0.3669 | 0.4032 | 0.4305 | 0.2154 | 0.2542 | 0.2474 | 0.3412 | 0.1250 | 0.1851 | 0.1407 | 0.3195 | 0.0747 | 0.1050 | 0.0932 | 0.1443 |
| EgoChoir [12] | 0.4959 | 0.6579 | 0.6591 | 0.6624 | 0.4302 | 0.5581 | 0.5744 | 0.5853 | 0.2612 | 0.3608 | 0.3711 | 0.3908 | 0.1987 | 0.2311 | 0.2445 | 0.2835 |
| Global Prior | 0.1359 | 0.2314 | 0.1878 | 0.3594 | 0.1048 | 0.1825 | 0.1324 | 0.3843 | 0.0331 | 0.0618 | 0.0359 | 0.3323 | 0.0146 | 0.0280 | 0.0149 | 0.3378 |
| Ours (w/o previous span) | 0.3424 | 0.4906 | 0.4864 | 0.5528 | 0.2616 | 0.3928 | 0.3891 | 0.4636 | 0.1071 | 0.1739 | 0.1742 | 0.2310 | 0.0594 | 0.0828 | 0.0896 | 0.0989 |
| Ours ($\mathcal{L}_{BCE}$) | 0.5730 | 0.7134 | 0.8234 | 0.6593 | 0.4594 | 0.6017 | 0.7580 | 0.5454 | 0.2836 | 0.3879 | 0.5853 | 0.3428 | 0.2059 | 0.2609 | 0.3618 | 0.2501 |
| Ours (Single-task) | 0.5832 | 0.7238 | 0.7751 | 0.7070 | 0.4721 | 0.6154 | 0.7195 | 0.5803 | 0.3351 | 0.4485 | 0.5292 | 0.4445 | 0.2494 | 0.3193 | 0.3958 | 0.3131 |
| Ours (w/o global embedding) | 0.5602 | 0.7042 | 0.7760 | 0.6779 | 0.4647 | 0.6128 | 0.6739 | 0.6169 | 0.3241 | 0.4476 | 0.5047 | 0.4756 | 0.2624 | 0.3487 | 0.3732 | 0.3903 |
| **Ours** (full) | 0.5838 | 0.7247 | 0.7848 | 0.7002 | 0.4886 | 0.6350 | 0.6883 | 0.6354 | 0.3513 | 0.4762 | 0.5242 | 0.5054 | 0.2836 | 0.3709 | 0.4088 | 0.4006 |

17 average, evaluating over the 2-second forecast windows. The resulting datasets comprise 23.2k
18 samples for FoVS-Aria and 341.4k samples for FoVS-EgoExo, providing rich visual span-based
19 forecasting samples across diverse scenarios, as illustrated in Fig. 6.

## A.2 Implementation Details

21 For all reported experiments, we use the Adam optimizer [1] with a learning rate of 1e-4, without
22 applying any scheduler or weight decay. We train our models with a batch size of 16 for 50 epochs
23 and select the epoch that achieves the highest average IoU across all visual spans on the validation
24 split for final testing. Input volumes are augmented through axis permutation and flipping (excluding
25 upside-down orientation), along with random translation applied up to 2 units.

26 The unidirectional transformer utilizes a feature dimension of $C = 1024$. To achieve this di-
27 mensionality, each U-Net layer reduces the feature dimension by a factor of 2, *i.e.*, an encoding
28 progression of 5-128-256-512-1024 for $16^3$ grids. Each U-Net layer consists of two applications of
29 Conv-BatchNorm-ReLU-Dropout blocks, with a dropout rate of 0.1.

30 While experimental results are reported for single runs, we conduct three random runs for our
31 method and observe performance variations of less than 1%, which proved negligible compared to
32 performance differences with other models. All experiments are conducted using NVIDIA RTX
33 A6000 GPUs with 48GB memory and 16 CPU cores. Additional implementation details can be found
34 in the source code.

## A.3 Environment and Asset Usage

36 The raw egocentric data sources used for curating the testbed are Aria Everyday Activities [2]
37 and Ego-Exo4D [3]. Aria Everyday Activities[1] is released under a custom license[2] that permits
38 academic research only, while Ego-Exo4D[3] uses a custom license[4] allowing both academic and
39 commercial usage. We confirm that our usage aligns with their intended purposes. Since both datasets
40 are collected using Aria glasses[5], we utilize the Project Aria Tools library[6] (Apache-2.0) for data
41 processing. Additionally, we conduct experiments using PyTorch 2.4.1, employing Open3D [4]
42 for data processing and the PyTorch-3DUNet [5] library for model construction, both of which are
43 distributed under the MIT License.

# B  Additional Experiments

## B.1 Additional Results on 3D Visual Span Forecasting

46 Table 5 presents the complete experimental results for FoVS-Aria, including both Precision and
47 Recall metrics. In some cases, the harmonic mean of Precision and Recall differs from the F1 score
48 because the harmonic mean was calculated at the sample level rather than globally, resulting in

---

[1] https://www.projectaria.com/datasets/aea/

[2] https://www.projectaria.com/datasets/aea/license/

[3] https://ego-exo4d-data.org/

[4] https://ego4d-data.org/pdfs/Ego-Exo4D-Model-License.pdf

[5] https://www.projectaria.com/

[6] https://github.com/facebookresearch/projectaria_tools

Table 6: Comparison of overlap-based metrics and saliency-based metrics on the FoVS-Aria test split.

| Ours | Orientation | | | | Peripheral$_{30°}$ | | | | Central$_{8°}$ | | | | Foveal$_{2°}$ | | | |
|---|---|---|---|---|---|---|---|---|---|---|---|---|---|---|---|---|
| | IoU | F1 | CC | AUC | IoU | F1 | CC | AUC | IoU | F1 | CC | AUC | IoU | F1 | CC | AUC |
| Single-task | 0.583 | 0.724 | 0.725 | 0.861 | 0.472 | 0.615 | 0.630 | 0.791 | 0.335 | 0.449 | 0.468 | 0.674 | 0.249 | 0.319 | 0.340 | 0.524 |
| w/o global | 0.560 | 0.704 | 0.711 | 0.847 | 0.465 | 0.613 | 0.631 | 0.827 | 0.324 | 0.448 | 0.477 | 0.765 | 0.262 | 0.349 | 0.376 | 0.613 |
| Full | 0.584 | 0.725 | 0.729 | 0.858 | 0.489 | 0.635 | 0.649 | 0.840 | 0.351 | 0.476 | 0.502 | 0.771 | 0.284 | 0.371 | 0.399 | 0.624 |

Table 7: SLAM sensitivity analysis on the FoVS-Aria test split.

| | Orientation | | Peripheral$_{30°}$ | | Central$_{8°}$ | | Foveal$_{2°}$ | |
|---|---|---|---|---|---|---|---|---|
| | IoU | F1 | IoU | F1 | IoU | F1 | IoU | F1 |
| EgoChoir [12] | 0.4959 | 0.6579 | 0.4302 | 0.5581 | 0.2612 | 0.3608 | 0.1987 | 0.2311 |
| Ours (original) | 0.5838 | 0.7247 | 0.4886 | 0.6350 | 0.3513 | 0.4762 | 0.2836 | 0.3709 |
| w/ temporal (5%) | 0.5592 | 0.6956 | 0.4666 | 0.6071 | 0.3343 | 0.4543 | 0.2697 | 0.3530 |
| w/ temporal (10%) | 0.5347 | 0.6674 | 0.4457 | 0.5810 | 0.3194 | 0.4344 | 0.2571 | 0.3371 |
| w/ translation (5cm) | 0.5814 | 0.7226 | 0.4871 | 0.6332 | 0.3399 | 0.4661 | 0.2611 | 0.3498 |
| w/ translation (10cm) | 0.5727 | 0.7158 | 0.4780 | 0.6258 | 0.3129 | 0.4407 | 0.2174 | 0.3022 |
| w/ rotation (2.5°) | 0.5818 | 0.7227 | 0.4882 | 0.6338 | 0.3461 | 0.4717 | 0.2681 | 0.3563 |
| w/ rotation (5°) | 0.5775 | 0.7193 | 0.4836 | 0.6298 | 0.3328 | 0.4575 | 0.2490 | 0.3357 |
| w/ Gaussian (2.5cm) | 0.5759 | 0.7190 | 0.4839 | 0.6313 | 0.3432 | 0.4697 | 0.2758 | 0.3661 |
| w/ Gaussian (5cm) | 0.5292 | 0.6831 | 0.4503 | 0.6037 | 0.3133 | 0.4436 | 0.2364 | 0.3336 |
| w/ point drop (10%) | 0.5823 | 0.7231 | 0.4890 | 0.6346 | 0.3500 | 0.4751 | 0.2823 | 0.3691 |
| w/ point drop (20%) | 0.5799 | 0.7212 | 0.4871 | 0.6330 | 0.3498 | 0.4741 | 0.2806 | 0.3676 |

different patterns of invalid values across metrics. For the 2D Gaze model, the numbers in gray (*i.e.*, orientation) are mechanically assigned based on camera position regardless of gaze anticipation quality, thereby identical for all cases. Examining Precision and Recall performance, our full model demonstrates superior results across all Foveal span metrics. However, for broader spans, single-task models and those using BCE loss achieve slightly higher precision scores. Nevertheless, these models tend to make relatively sparse predictions, resulting in notably lower recall performance. Consequently, Dice loss is preferred in our multi-task model due to its more balanced performance.

To provide complementary analysis on saliency metrics in our benchmarks, we report Correlation Coefficient (CC) and Area Under Curve (AUC) over the logit distribution of 3D grids using three variants of our framework in Table 6. We use AUC instead of KLD due to zero-value sensitivity of KLD [13] in our evaluation. The results demonstrate patterns that are generally consistent with previously reported metrics.

## B.2  Sensitivity Analysis on SLAM Keypoints

Since our approach utilizes semidense keypoints derived from SLAM, their quality could affect performance. However, this applies to any framework that relies on SLAM-derived keypoints as input. Still, it is feasible to utilize accurate pre-mapped information as a fallback option since our framework should typically be deployed in familiar everyday environments, *e.g.*, home and office.

To conduct a sensitivity analysis examining how the performance of our framework changes when SLAM struggles, we explore several types of sensory corruption across mild to severe scenarios. First, multi-sensor devices may experience temporary frame drops or time drift due to hardware issues, representing temporal corruption. Second, spatial degradation in egocentric localization may occur, which we apply separately to translation and rotation components. Finally, we consider corruption that can affect the set of keypoints by adding Gaussian noise to individual points or dropping certain points entirely.

Performance results measured on the FoVS-Aria test split for each corruption type are presented in Table 7. Under mild corruption, the impact on performance is negligible. However, applying higher-intensity corruption results in performance degradation of several percentage points. Among the different spans, those requiring wider coverage (Orientation and Peripheral) show smaller performance drops. In contrast, spans demanding higher precision (Central and Foveal) exhibit larger degradation. Despite these challenges, our framework maintains superior performance compared to EgoChoir even under severe corruption scenarios. This suggests that our framework can perform reliable forecasting despite some degree of SLAM-induced imprecision. In practice, real-world scenarios

| Previous Visual Span | CSTS | Ours | Ground Truth |
|---|---|---|---|

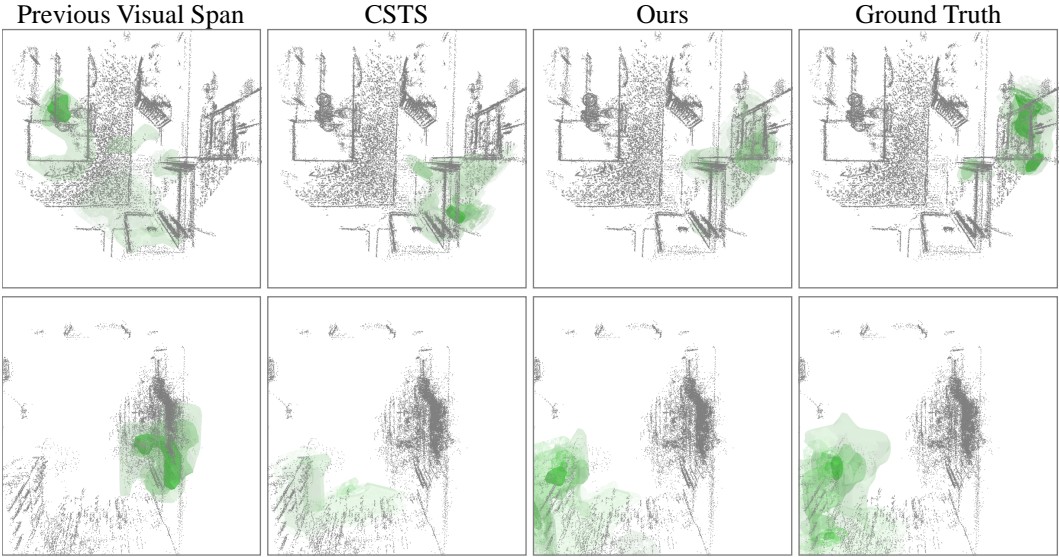

Figure 7: More qualitative examples on FoVS-Aria test split.

involve multiple types of corruption occurring in complex and somewhat unpredictable combinations. Therefore, training our framework to be robust against such corruption would represent an interesting extension for future work.

## B.3 More Qualitative Examples

Fig. 7 provides additional qualitative analysis of examples from FoVS-Aria. Consistent with the findings in Fig. 5, these results demonstrate our model's ability to perform effective forecasting across diverse scenarios and show meaningfully closer approximations to ground truth compared to other competitive models. Fig. 8 presents additional qualitative analysis of examples from FoVS-EgoExo. Our model performs well across various scenarios including bike repairing, cooking, and bouldering. For instance, it appropriately predicts the semantic significance of actions such as gathering items on a workbench or looking around while holding a knife in the kitchen, accurately anticipating the future visual focus these behaviors will lead to.

Fig. 9 shows qualitative examples of egocentric 2D gaze anticipation when our model's 3D inference results are projected to 2D. The results reveal that actual human gaze tends to be linked to specific 3D positions or objects rather than simply following head rotation patterns, which our model captures effectively. The final example illustrates a challenging case for our model in both 2D and 3D: when people interact with objects and create three-dimensional spaces that were not captured by semidense keypoints at previous time steps, performing precise inference becomes relatively difficult.

## C Limitations and Broader Impact

Our research interpret the primary source of perceptual intent in terms of gaze and peripheral span. However, human intent is comprehensively formed through the interplay of multiple sensory perceptions, including audio and proprioceptive inputs, beyond visual attention alone. Therefore, gaze alone may not fully capture the complexity of human intent. Future work could generalize toward forecasting human intent that can be identified through more broadly defined multisensory perception. Additionally, while we utilized semidense keypoints to minimize latency for real-time applications while conveying an accurate sense of distance, we did not interpret the visual span as a continuous surface representation. From this perspective, an approach that combines neural rendering [14] to represent visual span in a fine-grained manner would be beneficial for modeling intent with greater precision.

Moving forward, our research can be applied to facilitate service delivery in various egocentric latency-sensitive services by preemptively capturing the wearer's intent. For instance, it can predict

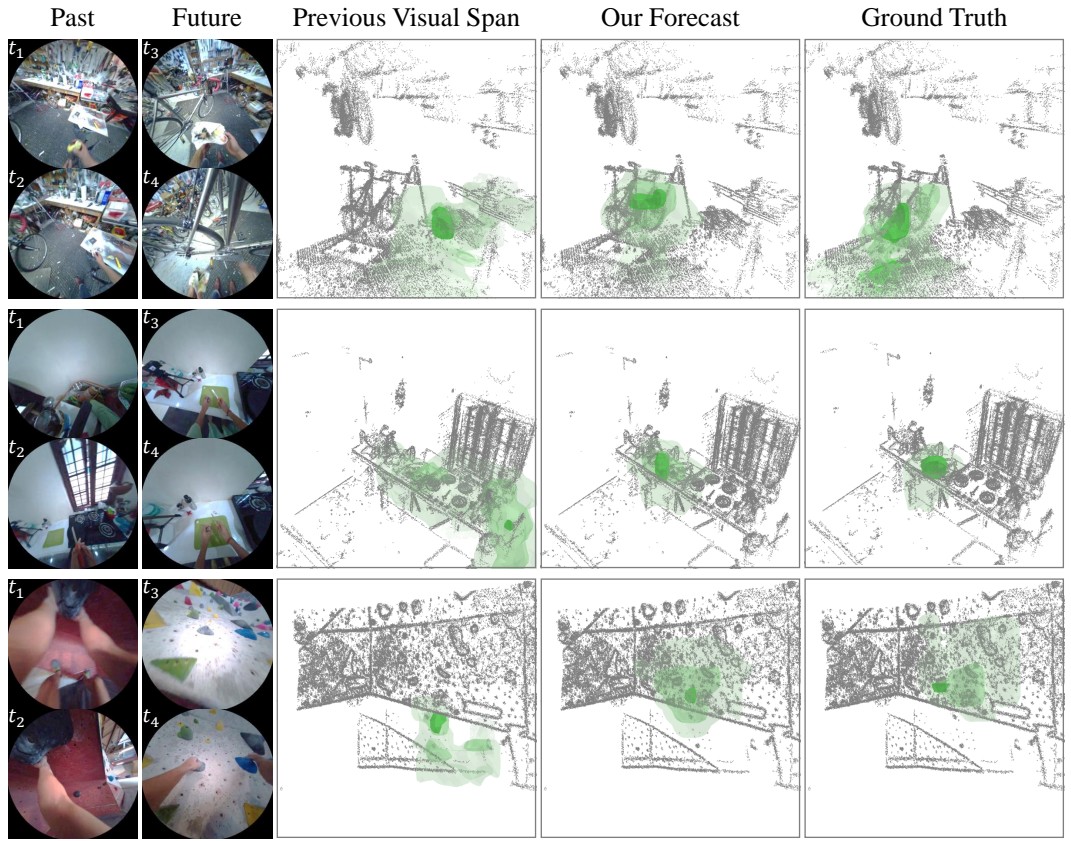

Figure 8: More qualitative examples on FoVS-EgoExo test split.

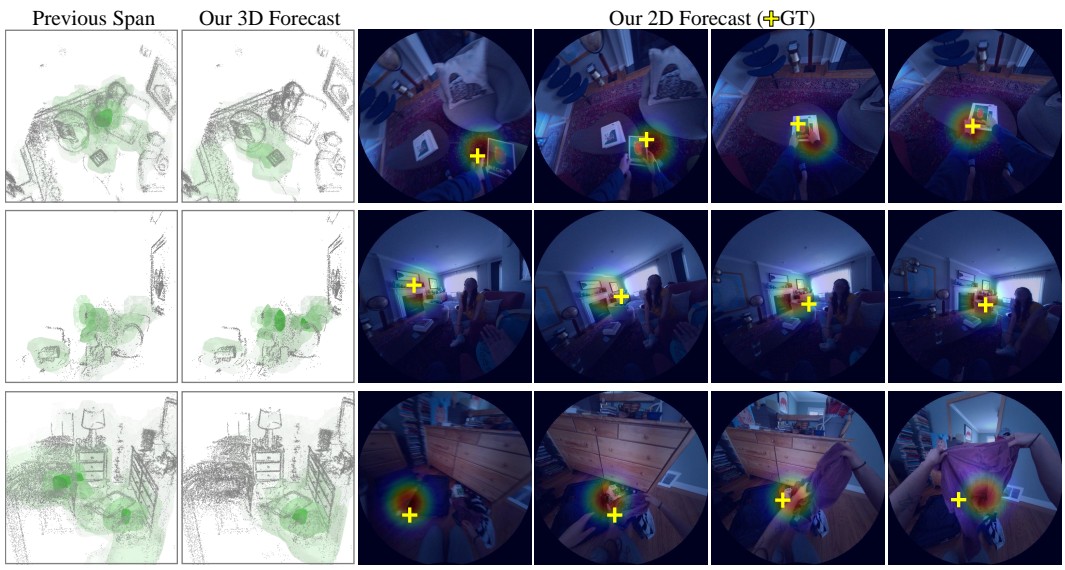

Figure 9: More qualitative examples on egocentric 2D gaze anticipation.

users' future focus or interest, enabling ambient computing to proactively adapt the surrounding environment in a more user-friendly manner. We anticipate that such technology will be useful for providing seamless access to desired objects or information for individuals with various impairments in a non-invasive manner. Furthermore, for general users, when providing augmented reality services, it will be possible to render or deliver information with higher fidelity in areas that align with user's intent.

On the other hand, using the wearer's perceptual input for model training or data utilization could potentially raise privacy issues. Our model is relatively unaffected by such concerns as it uses source data [2, 3] that are in compliance with privacy requirements and does not explicitly exploit personally identifiable information. However, careful consideration is needed when applying this technology in real-world scenarios. Since our methodology adopted a direct and lightweight approach in both volumetric region representation as visual span and network design due to latency-sensitive aspect of the problem setup, suggesting that on-device processing could also be a viable direction for mitigating privacy concerns.

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
