# OpenReview forum: "Gaze Beyond the Frame: Forecasting Egocentric 3D Visual Span"
_NeurIPS.cc/2025/Conference — NeurIPS 2025 spotlight_

### Official Review · Reviewer_7cjL · 2025-06-30

**Clarity:** 3
**Significance:** 3
**Originality:** 4
**Rating:** 5
**Confidence:** 3

**Summary:**

This paper introduces EgoSpanLift, a method that (1) lifts egocentric 2D gaze history into 3D volumetric “visual spans” via SLAM-derived keypoints, and (2) forecasts future spans using a hybrid 3D U-Net + transformer framework. To evaluate, the authors curate large-scale benchmarks: FoVS-Aria and FoVS-EgoExo to demonstrate substantial gains over adapted 2D gaze and 3D localization baselines across multiple metrics

**Questions:**

1. While the author says the latency introduced by the method is "trivial", I wonder if there is any data related to this? I would think any AR/VR application would be very sensitive to latency.

2. Some very minor formatting/grammar issues
- Binary cross-entropy and BCE should be associated at some point (maybe when it first appears in line 280), so that in Table 1, $L_{BCE}$ can be less confusing.
- In Fig. 1 and Fig. 3, Ground truth's truth is not capitalized, despite all other words being capitalized
- In Sec 5.1 and Sec 5.2, it should be "Forecasting", not "Forcasting"

**Ethical Concerns:**

["NO or VERY MINOR ethics concerns only"]

**Final Justification:**

I believe this paper is a positive contribution to the community. I will maintain my original positive rating.

**Limitations:**

Yes

**Quality:**

3

**Strengths And Weaknesses:**

# Strength
- Problem formulation is interesting and practical
- Writing about methodology is clear
- Methodology itself is solid, concise, with ample ablation study
- Benchmarking results are comprehensive, and the performance gain is substantial

# Weakness
- No obvious weakness
- However, I think the related work section can be enhanced.
    - https://arxiv.org/abs/2203.13116, https://arxiv.org/abs/2403.05046 addresses egocentric 3D user intention prediction
    - https://arxiv.org/abs/2307.08243 addresses 3D user hand prediction

---

> ### Author Rebuttal · Authors · 2025-07-31
>
> We thank reviewer 7cjL for the helpful feedback.
> To summarize all reviewer feedback, we are encouraged that the reviewers find our problem formulation interesting and advantageous (7cjL, LuMp, 23sp) while being methodologically solid and innovative (7cjL, LuMp) with clear writing (7cjL, k89X).
> Additionally, they find that our comprehensive experimental results demonstrate significant performance improvements (7cjL, LuMp, 23sp) with the curation of two new large-scale benchmarks (LuMp, 23sp).
> Below, we address the specific questions raised in reviewer 7cjL’s comments.
>
> ## (7cjL-Q1) Detailed analysis on the latency of our framework
>
> Our framework has two primary sources of latency: (i) extracting the relevant set of points from gaze and SLAM keypoints (performed every 100ms), and (ii) performing model inference every second on a set of points spanning two seconds.
>
> Since the point extraction can be pre-computed and stored at 10 fps for continuous use, we only need to consider the computation time for processing the final observation when calculating inference latency.
> We measured this in a resource-constrained environment compared to our training setup, using 8 CPU cores and a GPU with 12GB VRAM (e.g., Titan X). In the table below, all operations except model inference are processed on the CPU, with the following execution time:
>
> | **Gaze and keypoint processing**   |                 |
> |------------------------------------|-----------------|
> | Point preprocessing                | 4.541±1.999ms   |
> | 3D visual span localization        | 1.811±0.824ms   |
> | **From keypoints to model output** |                 |
> | Voxelization                       | 45.406±26.223ms |
> | Model inference                    | 19.483±8.234ms  |
> |                                    |                 |
> | **Average inference latency**      | 71.241ms        |
>
> The first stage, which handles outlier removal, axis-aligned bounding box cropping for keypoints, and selection of points within a certain degree of eccentricity from the gaze, can be processed within 10ms.
> In the second stage, we identified that the primary bottleneck lies in voxelization rather than in the model itself. This occurs because the large number of keypoints from the previous stage should be voxelized, whereas the model operates efficiently once it receives the 3D voxelized representation.
>
> Consequently, the average inference latency is 71.241ms, yielding a real-time factor of 0.036 when processing 2-second input for multi-second forecasting. This confirms our claim that the framework supports fast processing. However, actual AR/VR environments typically operate with even fewer computational resources, and thus additional optimization techniques such as model quantization or more efficient voxelization could be considered for further performance improvements.
>
> ## (7cjL-Q2) Suggestions on writing
>
> Thank you for your suggestion on the related work. We will include these papers and fix formatting issues.

---

> > ### Comment · Reviewer_7cjL · 2025-08-04
> >
> > I appreciate the author's effort in addressing my concerns. I will maintain my positive rating

---

> > > ### Author Response · Authors · 2025-08-05
> > >
> > > Thank you for your follow-up comment and for maintaining your positive evaluation. We are truly grateful for your time and effort in helping us improve the manuscript.

---

### Official Review · Reviewer_LuMp · 2025-06-30

**Clarity:** 3
**Significance:** 3
**Originality:** 3
**Rating:** 4
**Confidence:** 4

**Summary:**

This paper tackles a novel problem of egocentric 3D visual span forecasting. It introduces EgoSpanLift, a cone-based lifting mechanism that transforms 2D gaze data into 3D multi-level volumetric representations, and proposes a temporal forecasting model to predict future human visual attention in 3D space. The authors also curate two datasets for benchmarking this task. The proposed method demonstrates superior performance compared to existing baselines and shows promising potential for downstream applications in egocentric vision.

**Questions:**

1. The paper mentions real-time or latency-sensitive applications. Is the current model suitable for deployment on mobile or AR/VR hardware platforms?
2. Could the authors elaborate more on the practical applications of 3D visual span forecasting in the real world? What new possibilities does this 3D representation unlock that 2D gaze prediction cannot?
3. Why did you choose IoU and F1 as the primary evaluation metrics? Could commonly used 2D saliency metrics like KLdiv or CC also be applicable or provide complementary insights?
4. It might be interesting to consider subject-level information in future work, as different individuals can exhibit diverse gaze patterns. Incorporating such factors could potentially improve generalization across users.
5. In the broader impact section, the authors mention that this technology may help individuals with cognitive or perceptual impairments gain non-invasive access to desired objects or information. I would be curious to learn more about this point. Could the authors clarify which specific groups of users they are referring to, and how such support might be practically implemented?

Please refer to the above weaknesses and questions. If addressed appropriately, I would be open to raising my score.

**Ethical Concerns:**

["NO or VERY MINOR ethics concerns only"]

**Final Justification:**

Thank you for your detailed response, which has addressed most of my concerns. The computation time may still be insufficient to meet the demands of real-world applications, but this exceeds the scope of contribution for a single paper. I suggest that the authors (1) include a discussion on the primary evaluation metrics in the final version, as this is crucial for presenting a new paradigm, and (2) supplement the paper with more related works on egocentric and interaction-guided attention prediction, as the current version is lacking in this regard. I am willing to maintain my positive initial score.

**Limitations:**

Yes.

**Paper Formatting Concerns:**

None.

**Quality:**

3

**Strengths And Weaknesses:**

Strengths:
1. The paper introduces an interesting and timely new task that can significantly advance egocentric vision, especially in interaction-oriented settings.
2. The proposed 3D gaze representation and forecasting framework are innovative and clearly motivated by cognitive theories and real-world applications.

Weaknesses:
1. The evaluation on FoVS-EgoExo (Table 3) is somewhat limited. Unlike FoVS-Aria (Table 1), it lacks comprehensive results for extreme baselines such as Global Prior, 2D methods lifted via EgoSpanLift, and 3D localization models like OccFormer or VoxFormer. A fuller set of comparisons would strengthen the conclusions.
2. Several recent egocentric and interaction-guided attention prediction works are missing in the Related Work section. For example: CVPR'24 Learning from observer gaze: Zero-shot attention prediction oriented by human-object interaction recognition

---

> ### Author Rebuttal · Authors · 2025-07-31
>
> We thank reviewer LuMp for the constructive comments.
> To summarize all reviewer feedback, we are encouraged that the reviewers find our problem formulation interesting and advantageous (7cjL, LuMp, 23sp) while being methodologically solid and innovative (7cjL, LuMp) with clear writing (7cjL, k89X).
> Additionally, they find that our comprehensive experimental results demonstrate significant performance improvements (7cjL, LuMp, 23sp) with the curation of two new large-scale benchmarks (LuMp, 23sp).
> Below, we address the specific questions raised in reviewer LuMp's comments.
>
> ## (LuMp-Q1) Additional experiments on FoVS-EgoExo
> Due to space constraints, Table 3 previously reported only partial entries. We further report two additional ablation experiments, three 3D localization baselines, and one 2D gaze anticipation baseline:
> |           | O-IoU  | O-F1   | P-IoU  | P-F1   | C-IoU  | C-F1   | F-IoU  | F-F1   |
> |---------------------------|--------|--------|--------|--------|--------|--------|--------|--------|
> | CSTS + EgoSpanLift + [60] | -      | -      | 0.4978 | 0.6398 | 0.2867 | 0.4010 | 0.1556 | 0.2107 |
> | OccFormer                 | 0.1287 | 0.2280 | 0.0920 | 0.1685 | 0.0251 | 0.0490 | 0.0084 | 0.0167 |
> | VoxFormer                 | 0.1896 | 0.3188 | 0.1475 | 0.2571 | 0.0620 | 0.1168 | 0.0179 | 0.0350 |
> | EgoChoir                  | 0.3287 | 0.4948 | 0.2851 | 0.4437 | 0.1976 | 0.3300 | 0.1266 | 0.2247 |
> | Ours w/ BCEloss           | 0.5112 | 0.6621 | 0.4905 | 0.6338 | 0.3722 | 0.4867 | 0.2870 | 0.3578 |
> | Ours w/o global emb.      | 0.4998 | 0.6542 | 0.4892 | 0.6381 | 0.3954 | 0.5294 | 0.3475 | 0.4500 |
> | Ours                      | 0.5317 | 0.6827 | 0.5255 | 0.6694 | 0.4462 | 0.5730 | 0.3977 | 0.4932 |
>
> Overall, the results show a similar performance distribution to those reported on FoVS-Aria except for the 2D baseline often outperforming EgoChoir, presumably due to increased 3D scene complexity compared to FoVS-Aria. Due to time constraints, additional rows will be incorporated into the draft and the response once they become available.
>
> ## (LuMp-Q2) Detailed analysis on the latency of our framework
> Our framework has two primary sources of latency: (i) extracting the relevant set of points from gaze and SLAM keypoints (performed every 100ms), and (ii) performing model inference every second on a set of points spanning two seconds.
>
> Since the point extraction can be pre-computed and stored at 10 fps for continuous use, we only need to consider the computation time for processing the final observation when calculating inference latency.
> We measured this in a resource-constrained environment compared to our training setup, using 8 CPU cores and a GPU with 12GB VRAM (e.g., Titan X). In the table below, all operations except model inference are processed on the CPU, with the following execution time:
> | **Gaze and keypoint processing**   |                |
> |------------------------------------|-----------------|
> | Point preprocessing                | 4.541±1.999ms   |
> | 3D visual span localization        | 1.811±0.824ms   |
> | **From keypoints to model output** |                 |
> | Voxelization                       | 45.406±26.223ms |
> | Model inference                    | 19.483±8.234ms  |
> |                                    |                 |
> | **Average inference latency**      | 71.241ms        |
>
> The first stage, which handles outlier removal, axis-aligned bounding box cropping for keypoints, and selection of points within a certain degree of eccentricity from the gaze, can be processed within 10ms.
> In the second stage, we identified that the primary bottleneck lies in voxelization rather than in the model itself. This occurs because the large number of keypoints from the previous stage should be voxelized, whereas the model operates efficiently once it receives the 3D voxelized representation.
>
> Consequently, the average inference latency is 71.241ms, yielding a real-time factor of 0.036 when processing 2-second input for multi-second forecasting. This confirms our claim that the framework supports fast processing. However, actual AR/VR environments typically operate with even fewer computational resources, and thus additional optimization techniques such as model quantization or more efficient voxelization could be considered for further performance improvements.
>
> ## (LuMp-Q3) Practical applications of 3D visual span forecasting
> As our title "Gaze Beyond the Frame" implies, our 3D representation offers significant competitive advantages over 2D methods in the scenarios involving perception and interaction beyond the current visual frame. While 2D approaches are constrained to forecasting on existing visual frames, our method can predict attention patterns in previously unseen spatial regions.
> Foveated rendering in VR/MR systems exemplifies this advantage: when users turn their head to attend to different contexts, our 3D spatial awareness maintains consistent predictions across head movements and viewpoint changes, whereas 2D methods struggle with such dynamic scenarios.
>
> Moreover, our approach potentially enables various forms of proactive environmental assistance by integrating ambient intelligence. For instance, systems can adjust lighting or display configurations in spaces where users are predicted to look, facilitating faster information perception. Additionally, by considering users' established focusing patterns, ambient assistants can proactively remind users of information they might otherwise overlook, creating more intuitive human-environment interactions.
>
> Temporally persistent 3D representations offer another key advantage. While 2D-based forecasting becomes less reliable as self-motion increases, our framework utilizes 3D spatial representation that is relatively robust to self-motion. This opens up the potential for extending our framework for forecasting on a longer horizon, which could enable wearable devices to learn and anticipate user behaviors in their daily 3D environments rather than being limited to momentary 2D observations.
>
> Lastly, our experimental results demonstrate that 3D visual span predictions, when back-projected to 2D image planes, achieve performance comparable to 2D-specific models without requiring 2D-specific training. This suggests that our 3D approach not only unlocks new capabilities but also maintains the effectiveness of traditional 2D methods while providing the additional spatial understanding necessary for future egocentric applications.
>
> ## (LuMp-Q4) Rationale behind our primary evaluation metrics
> We primarily build upon the recent 2D egocentric gaze estimation and anticipation literature [17,33,34], where F1, Precision, and Recall are well-established metric choices. We further complement these metrics by incorporating measures that reflect the nature of 3D forecasting: IoU for measuring overlaps between volumetric regions and metric distance (Table 2) to compensate for narrow regions from the foveal span.
>
> To provide complementary analysis on saliency metrics in our benchmarks, we report CC and AUC over the logit distribution of 3D grids using three variants of our framework. Note that AUC is reported instead of KLD due to zero-value sensitivity of KLD [TPAMI’19] observed in our evaluation:
> |       | O-IoU  | O-F1   | O-CC   | O-AUC  | P-IoU  | P-F1   | P-CC   | P-AUC  |
> |-------------|--------|--------|--------|--------|--------|--------|--------|--------|
> | Single-task | 0.5831 | 0.7237 | 0.7253 | 0.8610 | 0.4721 | 0.6154 | 0.6295 | 0.7913 |
> | w/o Global  | 0.5602 | 0.7042 | 0.7113 | 0.8470 | 0.4647 | 0.6128 | 0.6308 | 0.8265 |
> | Ours        | 0.5838 | 0.7247 | 0.7293 | 0.8582 | 0.4886 | 0.6350 | 0.6492 | 0.8402 |
> |             | C-IoU  | C-F1   | C-CC   | C-AUC  | F-IoU  | F-F1   | F-CC   | F-AUC  |
> | Single-task | 0.3351 | 0.4485 | 0.4679 | 0.6736 | 0.2494 | 0.3193 | 0.3398 | 0.5240 |
> | w/o Global  | 0.3242 | 0.4476 | 0.4767 | 0.7651 | 0.2624 | 0.3487 | 0.3755 | 0.6132 |
> | Ours        | 0.3513 | 0.4762 | 0.5020 | 0.7714 | 0.2836 | 0.3709 | 0.3990 | 0.6242 |
>
> The results demonstrate patterns that are generally consistent with previously reported metrics. We will include this additional analysis in the Appendix.
>
> ## (LuMp-Q5) Elaboration on users with cognitive and perceptual impairments
> Several user groups with cognitive and perceptual impairments could potentially benefit from this technology. These include individuals with visual perceptual deficits who may miss objects in their peripheral vision, those with age-related visual degeneration who experience slower or less accurate visual targeting compared to typical users, and people with mild cognitive impairment who could benefit from contextual reminders about their behavior sequences and task progression.
>
> The implementation of assistive support for these users would vary significantly depending on the underlying device or platform. For instance, in display-equipped systems, highlighted overlays could be used to draw attention to forecasted visual spans, helping users identify areas of upcoming interaction. In environments with voice assistants or ambient intelligence systems, direct or indirect cues (e.g., haptic or voice assistant notification) could be provided to users regarding perceptually overlooked contexts, thereby facilitating both perception and interaction. Such proactive assistance has the potential to reduce cognitive load while maintaining user autonomy by anticipating user needs based on their natural gaze patterns and environmental context.
>
> ## (LuMp-Q6) Suggestions on future work and relevant papers
> Thank you for your suggestion on the related work and future direction. Enhancing subject-aware personalization of 3D forecasting could be an interesting extension by incorporating methods like test-time adaptation. We will discuss them in our final draft.
>
> ## Reference
> [TPAMI’19] Bylinskii et al. (2019). What Do Different Evaluation Metrics Tell Us about Saliency Models?

---

> > ### Comment · Area_Chair_SPNw · 2025-08-05
> >
> > LuMp, please could you take a look at the author response above and whether it addresses any remaining concerns you have, e.g. limited evaluation

---

> > ### Comment · Reviewer_LuMp · 2025-08-06
> >
> > Thank you for your detailed response, which has addressed most of my concerns. The computation time may still be insufficient to meet the demands of real-world applications, but this exceeds the scope of contribution for a single paper. I suggest that the authors (1) include a discussion on the primary evaluation metrics in the final version, as this is crucial for presenting a new paradigm, and (2) supplement the paper with more related works on egocentric and interaction-guided attention prediction, as the current version is lacking in this regard.

---

> > > ### Author Response · Authors · 2025-08-07
> > >
> > > Thank you for your additional feedback, and we are delighted to hear that our responses effectively addressed your concerns. While our experimental results in LuMp-Q2 demonstrate the capability for real-time processing (10+ inference per second), tackling the challenges associated with real-world deployment could be an interesting avenue for future extension. We have incorporated the clarification regarding our choice of metrics and expanded the related work section with additional comprehensive coverage. We will ensure that these improvements are well-reflected in the final draft.

---

### Official Review · Reviewer_23sp · 2025-07-02

**Clarity:** 3
**Significance:** 2
**Originality:** 3
**Rating:** 5
**Confidence:** 4

**Summary:**

This paper presents a new way to predict where a person will look next, shifting the problem from a 2D image to a 3D environment. The authors argue that this is more robust/fundamental because people naturally direct their attention to objects and locations in 3D space. Their solution has two main parts:
- The first, EgoSpanLift, takes the video from a user's perspective and uses SLAM to build a 3D map of the surroundings in real-time. Then, using the user's gaze direction, it performs a simple geometric calculation to determine which 3D points fall within different levels of their visual span (ranging from a tiny foveal focus to a much wider peripheral one). The result is a set of 3D grids that represent where the user's attention is focused at any given moment.
- The second is a forecasting network designed to predict what these 3D attention grids will look like in the near future. It takes a history of the grids created by EgoSpanLift and feeds them into a model that uses a 3D U-Net to understand the spatial patterns of attention and a transformer to understand how that attention moves over time. The final output is a predicted 3D volumetric region where the user's visual perception will focus next. To prove their method works, the authors created two new large-scale benchmarks from existing egocentric video datasets and showed that their approach outperforms a wide range of other methods.

**Questions:**

Please address my comments under Weaknesses.

**Ethical Concerns:**

["NO or VERY MINOR ethics concerns only"]

**Final Justification:**

I have read the rebuttal carefully and decided to update my review -- same rating (5/accept) but with a higher confidence (4).

**Paper Formatting Concerns:**

None.

**Quality:**

3

**Strengths And Weaknesses:**

**Strengths**:
- IMO the paper frames the problem of visual attention/gaze estimation in a sensible fashion. By constructing a state of the 3D environment (via SLAM) and modeling the user's focus here, this formulation provides many advantages (e.g., disentanglement from the user's own motion, 3D is more fundamental than 2D).
- The authors went through the process of curating two large-scale benchmarks, FoVS-Aria and FoVS-EgoExo, specifically for this new 3D forecasting problem.
- They show that their framework significantly outperforms a wide range of other approaches, including models adapted from both 2D gaze prediction and other 3D localization tasks. The improvement is especially large when predicting the user's precise/foveal gaze.

**Weaknesses**:
- The proposed system relies on getting a good 3D map from SLAM. The authors harness "3D semidense keypoints" from SLAM, but I'm unclear how important it is / what happens when the SLAM algorithm struggles (e.g. during fast motion)  such that a good map is not produced. If the foundational map is inaccurate, the entire gaze forecasting pipeline will be thrown off, limiting its reliability in the real world.
- It might be too computationally heavy for real-world use right now, and it would be nice if the authors can shed some light on this aspect. The forecasting network uses a 3D U-Net and a unidirectional transformer on a sequence of 3D volumetric grids. This kind of architecture is powerful but also known to be computationally hungry. For latency-sensitive applications like AR/VR running on a mobile headset, this could be too slow.

---

> ### Author Rebuttal · Authors · 2025-07-31
>
> We thank reviewer 23sp for the constructive feedback.
> To summarize all reviewer feedback, we are encouraged that the reviewers find our problem formulation interesting and advantageous (7cjL, LuMp, 23sp) while being methodologically solid and innovative (7cjL, LuMp) with clear writing (7cjL, k89X).
> Additionally, they find that our comprehensive experimental results demonstrate significant performance improvements (7cjL, LuMp, 23sp) with the curation of two new large-scale benchmarks (LuMp, 23sp).
> Below, we address the specific questions raised in reviewer 23sp's comments.
>
> ## (23sp-Q1) What happens when SLAM struggles?
> Since our approach utilizes semidense keypoints derived from SLAM, their quality could affect performance. However, this applies to any framework that relies on SLAM-derived keypoints as input. Still, it is feasible to utilize accurate pre-mapped information as a fallback option since our framework should typically be deployed in familiar everyday environments, e.g., home and office.
>
> To conduct a sensitivity analysis examining how the performance of our framework changes when SLAM struggles, we explore several types of sensory corruption across mild to severe scenarios. First, multi-sensor devices may experience temporary frame drops or time drift due to hardware issues, representing temporal corruption. Second, spatial degradation in egocentric localization may occur, which we apply separately to translation and rotation components. Finally, we consider corruption that can affect the set of keypoints by adding Gaussian noise to individual points or dropping certain points entirely.
> Performance results measured on the FoVS-Aria test split for each corruption type are presented in the following table:
>
> |                     | O-IoU  | O-F1   | P-IoU  | P-F1   | C-IoU  | C-F1   | F-IoU  | F-F1   |
> |---------------------|--------|--------|--------|--------|--------|--------|--------|--------|
> | Baseline (EgoChoir) | 0.4959 | 0.6579 | 0.4302 | 0.5581 | 0.2612 | 0.3608 | 0.1987 | 0.2311 |
> | Ours                | 0.5838 | 0.7247 | 0.4886 | 0.6350 | 0.3513 | 0.4762 | 0.2836 | 0.3709 |
> | Temporal (5%)       | 0.5592 | 0.6956 | 0.4666 | 0.6071 | 0.3343 | 0.4543 | 0.2697 | 0.3530 |
> | Temporal (10%)      | 0.5347 | 0.6674 | 0.4457 | 0.5810 | 0.3194 | 0.4344 | 0.2571 | 0.3371 |
> | Translation (5cm)   | 0.5814 | 0.7226 | 0.4871 | 0.6332 | 0.3399 | 0.4661 | 0.2611 | 0.3498 |
> | Translation (10cm)  | 0.5727 | 0.7158 | 0.4780 | 0.6258 | 0.3129 | 0.4407 | 0.2174 | 0.3022 |
> | Rotation (2.5°)     | 0.5818 | 0.7227 | 0.4882 | 0.6338 | 0.3461 | 0.4717 | 0.2681 | 0.3563 |
> | Rotation (5°)       | 0.5775 | 0.7193 | 0.4836 | 0.6298 | 0.3328 | 0.4575 | 0.2490 | 0.3357 |
> | Gaussian (2.5cm)    | 0.5759 | 0.7190 | 0.4839 | 0.6313 | 0.3432 | 0.4697 | 0.2758 | 0.3661 |
> | Gaussian (5cm)      | 0.5292 | 0.6831 | 0.4503 | 0.6037 | 0.3133 | 0.4436 | 0.2364 | 0.3336 |
> | Point drop (10%)    | 0.5823 | 0.7231 | 0.4890 | 0.6346 | 0.3500 | 0.4751 | 0.2823 | 0.3691 |
> | Point drop (20%)    | 0.5799 | 0.7212 | 0.4871 | 0.6330 | 0.3498 | 0.4741 | 0.2806 | 0.3676 |
>
> Under mild corruption, the impact on performance is negligible. However, applying higher-intensity corruption results in performance degradation of several percentage points. Among the different spans, those requiring wider coverage (Orientation and Peripheral) show smaller performance drops. In contrast, spans demanding higher precision (Central and Foveal) exhibit larger degradation. Despite these challenges, our framework maintains superior performance compared to EgoChoir even under severe corruption scenarios.
> This suggests that our framework can perform reliable forecasting despite some degree of SLAM-induced imprecision. In practice, real-world scenarios involve multiple types of corruption occurring in complex and somewhat unpredictable combinations. Therefore, training our framework to be robust against such corruption would represent an interesting extension for future work.
>
> ## (23sp-Q2) Detailed analysis on the latency of our framework
>
> Our framework has two primary sources of latency: (i) extracting the relevant set of points from gaze and SLAM keypoints (performed every 100ms), and (ii) performing model inference every second on a set of points spanning two seconds.
>
> Since the point extraction can be pre-computed and stored at 10 fps for continuous use, we only need to consider the computation time for processing the final observation when calculating inference latency.
> We measured this in a resource-constrained environment compared to our training setup, using 8 CPU cores and a GPU with 12GB VRAM (e.g., Titan X). In the table below, all operations except model inference are processed on the CPU, with the following execution time:
>
> | **Gaze and keypoint processing**   |                 |
> |------------------------------------|-----------------|
> | Point preprocessing                | 4.541±1.999ms   |
> | 3D visual span localization        | 1.811±0.824ms   |
> | **From keypoints to model output** |                 |
> | Voxelization                       | 45.406±26.223ms |
> | Model inference                    | 19.483±8.234ms  |
> |                                    |                 |
> | **Average inference latency**      | 71.241ms        |
>
> The first stage, which handles outlier removal, axis-aligned bounding box cropping for keypoints, and selection of points within a certain degree of eccentricity from the gaze, can be processed within 10ms.
> In the second stage, we identified that the primary bottleneck lies in voxelization rather than in the model itself. This occurs because the large number of keypoints from the previous stage should be voxelized, whereas the model operates efficiently once it receives the 3D voxelized representation.
>
> Consequently, the average inference latency is 71.241ms, yielding a real-time factor of 0.036 when processing 2-second input for multi-second forecasting. This confirms our claim that the framework supports fast processing. However, actual AR/VR environments typically operate with even fewer computational resources, and thus additional optimization techniques such as model quantization or more efficient voxelization could be considered for further performance improvements.

---

> > ### Author Response · Authors · 2025-08-08
> >
> > Dear Reviewer 23sp,
> >
> > We sincerely appreciate your insightful feedback on our manuscript. With the reviewer-author discussion phase nearing its conclusion, we wanted to follow up to see if our response has satisfactorily resolved the issues you raised. Should you have any additional observations or queries, we would be happy to address them.

---

> ### Comment · Area_Chair_SPNw · 2025-08-05
>
> 23sp, please could you take a look at the author response above and whether it addresses any remaining concerns you have, e.g. the requirement of a good 3D SLAM map.

---

### Official Review · Reviewer_k89X · 2025-07-03

**Clarity:** 2
**Significance:** 3
**Originality:** 2
**Rating:** 4
**Confidence:** 4

**Summary:**

The authors propose an approach to forecast "visual spans" form egocentric recordings. The method predicts in 3D, by lifting from 2D. The method is evaluated both in the 3D scenario, as well as for 2D egocentric gaze anticipation.

**Questions:**

To re-consider my score, I would like to see a convincing clarification of the concept of "visual span" (and how it relates to and motivates the author's works), and a convincing argument why the problem can not simply be addressed by doing 3D gaze anticipation + post-processing.

**Ethical Concerns:**

["NO or VERY MINOR ethics concerns only"]

**Final Justification:**

The author's promise to clarify the concept of visual span and the additional evaluations lead me to increase my score. I am still not totally convinced about where the difference to the postprocessing approach really lies. It would be great to convey a better intuition on this. I recommend not to use the name "visual span", as this terminology is too messy in the context of the author's work and will further confuse readers.

**Limitations:**

Limitations should be discussed in the main paper.

**Paper Formatting Concerns:**

no major issues

**Quality:**

2

**Strengths And Weaknesses:**

Strengths & Weaknesses:
-----------------------

The problem is relevant and the paper quite readable. The technical description appears to be valid, and is supported by adequate figures. My main criticisms are concerning the conceptual motivation/background of the work and an inadequate representation of previous works. I am also questioning whether "egocentric span prediction" as formulated by the authors has relevance on top of the task of 3D gaze anticipation.

Concept from visual attention research are used in an inconsistent and confusing way. Sometimes authors talk about forecasting gaze, sometimes about predicting future visual focus, most prominently (and in the title of the method), authors talk about forecasting future visual spans.

Concerning the terminology of "visual span": Visual span is mainly used in the context of reading and denotes the number of letters to the left and right of the fixation point that can be recognised correctly. See Frey & Bosse (2018) for a clarification on the terminology. The visual span depends on different aspects of the stimulus, e.g. character orientation or crowding. The crucial point here is that the visual span is not a cone around the gaze point that is independent of the stimulus. In contrast, the authors in the present paper do not capture visual span, as they simply define a stimulus-independent stack of cones around the gaze point. They also do not evaluate visual span prediction. To evaluate visual span, they would need to check whether their method is able to predict which objects around the gaze point can be recognised by the user.

I am not convinced that defining these regions around the gaze point gives us any new insights in the evaluation. Alternatively, the authors could simply evaluate the accuracy of 3D gaze prediction. My view is supported by Table 1 in the paper: the ordering of methods is highly consistent between Orientation metrics and Peripheral/Central/Foveal metrics. The same applies for Table 3. An alternative to the way the authors pose the problem is to simply anticipate 3D gaze, evaluate the quality of this gaze, and provide peripheral/central/foveal areas as a post-processing step.

In line 32, the authors state that "However, attempts to forecast human visual perception itself remain less explored."
I am not sure about this statement, as there are already quite a few works that try to do this (and are not referenced by the authors), e.g. see Steil et al. (2018), Hu et al. (2021), Rolff et al. (2022).
A similar point concerns lines 40-42: "While previous research has shown impressive results in predicting egocentric future gaze fixations on 2D image frames [16, 17], forecasting gaze for dynamic scenarios in 2D remains unclear.

The related work section is also misleading in these respects. For example, in the paragraph on "Egocentric Gaze Prediction", authors refer to works such as gaze following ([24, 25]) which is not an egocentric setting, and also not about anticipation. Relevant gaze anticipation works are not discussed (e.g. Steil et al., 2018; Hu et al., 2021; Rolff et al., 2022).



References:
-----------
- Frey, A., & Bosse, M. L. (2018). Perceptual span, visual span, and visual attention span: Three potential ways to quantify limits on visual processing during reading. Visual Cognition, 26(6), 412-429.
- Steil, J., Müller, P., Sugano, Y., & Bulling, A. (2018, September). Forecasting user attention during everyday mobile interactions using device-integrated and wearable sensors. In Proceedings of the 20th international conference on human-computer interaction with mobile devices and services (pp. 1-13).
- Hu, Z., Bulling, A., Li, S., & Wang, G. (2021). Fixationnet: Forecasting eye fixations in task-oriented virtual environments. IEEE Transactions on Visualization and Computer Graphics, 27(5), 2681-2690.
- Rolff, T., Harms, H. M., Steinicke, F., & Frintrop, S. (2022, September). Gazetransformer: Gaze forecasting for virtual reality using transformer networks. In DAGM German Conference on Pattern Recognition (pp. 577-593). Cham: Springer International Publishing.

---

> ### Author Rebuttal · Authors · 2025-07-31
>
> We thank reviewer k89X for the detailed comments.
> To summarize all reviewer feedback, we are encouraged that the reviewers find our problem formulation interesting and advantageous (7cjL, LuMp, 23sp) while being methodologically solid and innovative (7cjL, LuMp) with clear writing (7cjL, k89X).
> Additionally, they find that our comprehensive experimental results demonstrate significant performance improvements (7cjL, LuMp, 23sp) with the curation of two new large-scale benchmarks (LuMp, 23sp).
> Below, we address the specific questions raised in reviewer k89X’s comments.
>
> ## (k89X-Q1) Conceptual motivation of our work
>
> As noted in L39-40, we acknowledge that the term 'visual span' originates largely from text and symbolic reading research in cognitive psychology. While we draw inspiration from this foundation, our work addresses daily and casual behaviors and interactions from an egocentric perspective, without aiming to provide precise computational modeling of the original terminology.
> While reading-based visual span depends on specific textual stimuli, forecasting visual perception in egocentric daily activities involve diverse, dynamic environments where stimulus-independent spatial regions provide a more practical and generalizable framework. We will clarify the terminology discrepancies in our final draft.
> Please note that the other three reviewers have recognized the merit of this conceptual formulation, with reviewer 7cjL highlighting our 'practical problem formulation', reviewer LuMp noting that we are 'clearly motivated by cognitive theories and real-world applications', and reviewer 23sp stating that we 'frame the problem in a sensible fashion that provides many advantages,' respectively.
>
> Our cone-based multi-level interpretation serves as a plausible and practical representation for modeling eccentricity and periphery, which is reflected in our substantial performance gains over prior art. We note that gaze, visual focus, and visual span are known to be interconnected [VisCog’13, TPAMI’18, NatHumBehav’19], and similar concepts and wordings are commonly used in related computer vision work, including previous research [16, 17, 32, 33, 34, ETRA’10, ICCV’21].
> Importantly, our core methodological and empirical contributions, which have been appreciated by the other three reviewers, remain valuable regardless of naming choices. To address any remaining terminological concerns, we are open to adopting alternative terms.
>
> ## (k89X-Q2) Rationale behind our multi-level representation
>
> While the monotonic results in Table 1 across different spans may appear redundant, this impression arises simply from the broad prediction target regions. In reality, multi-level interpretation carries significant implications due to uncertainties in self-motion forecasting and geometric correspondence between 2D and 3D spaces. To demonstrate this importance, we conducted an ablation study applying post-processing to 3D gaze forecasting using a model trained with the multi-level components removed from our framework:
>
> |                          | P-IoU  | P-F1   | C-IoU  | C-F1   | F-IoU  | F-F1   | 2D-F1 | 2D-Pr | 2D-Re |
> |--------------------------|--------|--------|--------|--------|--------|--------|-------|-------|-------|
> | 2D gaze + postprocessing | 0.4567 | 0.6076 | 0.2342 | 0.3423 | 0.1388 | 0.1948 | 0.515 | 0.497 | 0.535 |
> | 3D gaze + postprocessing | 0.4723 | 0.6214 | 0.2666 | 0.3791 | 0.2494 | 0.3193 | 0.505 | 0.432 | 0.608 |
> | Ours (Single-task)       | 0.4721 | 0.6154 | 0.3351 | 0.4485 | 0.2494 | 0.3193 | 0.505 | 0.432 | 0.608 |
> | Ours                     | 0.4886 | 0.6350 | 0.3513 | 0.4762 | 0.2836 | 0.3709 | 0.515 | 0.440 | 0.619 |
>
> Predicting only gaze and applying post-processing is clearly suboptimal. Compared to the foveal forecast performance when jointly predicting all multi-level spans, foveal forecast performance shows an F1 decrease of approximately 5.2, stressing the importance of our multi-level representation and multi-task objectives.
> Furthermore, the central forecast exhibits an F1 decrease of approximately 9.7, which is substantially larger than the 2.8 F1 performance decrease observed in our model trained to predict only the central span. This demonstrates that predictive performance across different spans involves complex geometric and learning dynamics that extend beyond simple monotonic relationships.
> We will include this analysis in our final draft to clarify the benefit of our formulation.
>
> ## (k89X-Q3) Suggestions on relevant papers
>
> Thank you for your suggestion regarding the related work from mobile and VR research. We will cite these works [MobileHCI'18, TVCG'21, GCPR'22] and revise our discussion accordingly.
> Please note that we reference works [24, 25] in our related work section before narrowing our scope to egocentricity, as these are some of the seminal works for gaze comprehension in computer vision to the best of our knowledge.
>
> ## Reference
>
> [ETRA’10] Blignaut (2010). Visual Span and Other Parameters for the Generation of Heatmaps.
>
> [VisCog’13] Nuthmann (2013). On the Visual Span during Object Search in Real-world Scenes.
>
> [TPAMI’18] Masse et al. (2018). Tracking Gaze and Visual Focus of Attention of People Involved in Social Interaction.
>
> [NatHumBehav’19] van Ede et al. (2019). Human Gaze Tracks Attentional Focusing in Memorized Visual Space.
>
> [ICCV’21] Jiang and Ithapu (2021). Egocentric Pose Estimation from Human Vision Span.

---

> > ### Comment · Reviewer_k89X · 2025-08-04
> >
> > The author's promise to clarify the concept of visual span and the additional evaluations lead me to increase my score. I am still not totally convinced about where the difference to the postprocessing approach really lies. It would be great to convey a better intuition on this.
> > I recommend not to use the name "visual span", as this terminology is too messy in the context of the author's work and will further confuse readers.

---

> > > ### Author Response · Authors · 2025-08-05
> > >
> > > Thank you for taking the time to review our rebuttal. We are very pleased that our clarification of our conceptual motivation and the proposed framework was helpful in addressing your concerns.
> > >
> > > Beyond the substantial experimental gains (+9.7 C-F1 in k89X-Q2), two key conceptual differences between our framework and the postprocessing variant are (i) learning from multi-level representation and (ii) mitigating uncertainties in self-motion anticipation through end-to-end prediction.
> > > Leveraging the interconnection between gaze and periphery [15] allows us to capture cues about future gaze from previous periphery and forecast future periphery in light of previous gaze, which can be observed in several qualitative examples.
> > >
> > > Moreover, given the nature of the viewing frustum, extending 3D gaze predictions to broader spans necessitates the forecasting of egocentric 6DoF pose trajectories with a separate postprocessing stage, which propagates `uncertainties in self-motion forecasting and geometric correspondence between 2D and 3D spaces,` e.g., (23sp-Q1).
> > > In contrast, our end-to-end framework does not assume a specific forecasted trajectory and predicts plausible 3D spans within the scene while implicitly learning to mitigate such uncertainties.
> > >
> > > We will incorporate your suggestions in our final draft.

---

### Note · Authors · 2025-08-16

We sincerely thank the reviewers, ACs, and SACs for their effort and constructive feedback on our paper.
To summarize, our research on forecasting the _gaze beyond the frame_ provides the following core contributions:
- We addressed the novel challenge of egocentric 3D visual span forecasting to anticipate where a person's visual perception will be focused in the surroundings.
- The proposed framework comprises two key components: (1) EgoSpanLift for lifting 2D gazes and spans to 3D regions and (2) a framework for effectively forecasting future 3D visual spans.
- We rigorously curated two large-scale benchmarks from egocentric multisensory data, where our framework outperformed a wide array of prior art by a large margin.
- When back-projected to 2D, our 3D predictions achieve on-par performance with state-of-the-art 2D methods without 2D-specific training.

In the pre-rebuttal evaluation, the reviewers appreciated our problem formulation as interesting and advantageous (7cjL, LuMp, 23sp) while recognizing the soundness and innovation of our framework (7cjL, LuMp) along with clear presentation (7cjL, k89X). They also acknowledged that our extensive experimental results demonstrate substantial performance gains (7cjL, LuMp, 23sp) with two new large-scale benchmarks (LuMp, 23sp).

In response to the reviewers' feedback, we have addressed the following concerns, with positive reviewer responses including an increase in final rating:
- We decomposed the latency of our framework with a resource-constrained setup, which demonstrates real-time (10+ fps) processing capability.
- We strengthened experimental analysis by incorporating additional baselines and ablation studies on FoVS-EgoExo and computing saliency-based metrics.
- We explained the practical benefits of our framework compared to 2D-based frameworks and for users with impairments.
- We conducted a sensitivity analysis of the model with respect to diverse types of SLAM inaccuracies, where our model still outperformed all prior arts.
- We clarified the conceptual motivation of our work in light of our practically oriented problem formulation and previous research covering similar concepts.
- We provided conceptual and experimental analysis of the strengths of our proposed multi-level representation over other conceivable design choices.

We will ensure that all discussion points are properly integrated into the final draft and will publicly release the source code for our framework and benchmarks.

---

### Decision · Program_Chairs · 2025-09-17

**Decision:**

Accept (spotlight)

**Comment:**

This paper received 2 borderline accept scores and 2 accept scores as final ratings. Initially, reviewers liked the paper due to the curation of two datasets; strong motivations; interesting and timeliness of the proposed new task; and clear writing within the paper. They also had some criticisms towards the paper, namely that the concept of a visual span and how it could be solved in other ways; whether the model is too heavy for real-time applications; the underlying need for a strong SLAM map; and some missing related works. During the rebuttal stage, many of the reviewers' questions were answered with new results; additional descriptions; and analysis. Accordingly, all reviewers gave a positive final rating towards the paper. The AC agrees with the scores of the reviewers and sees no need to overturn the decision and recommends acceptance for the paper.

The AC reminds the authors that they should update the camera ready with the suggestions from the reviewers and promises that they made during the rebuttal stage.